# Discretization-free Multicalibration through Loss Minimization over Tree Ensembles

**Hongyi Henry Jin**[*]
University of California, Los Angeles
Los Angeles, CA, USA

**Zijun Ding**
Carnegie Mellon University
Pittsburgh, PA, USA

**Dung Daniel Ngo**[*]
J.P. Morgan Chase AI Research
New York, NY, USA

**Zhiwei Steven Wu**
Carnegie Mellon University
Pittsburgh, PA, USA

## Abstract

In recent years, multicalibration has emerged as a desirable learning objective for ensuring that a predictor is calibrated across a rich collection of overlapping subpopulations. Existing approaches typically achieve multicalibration by discretizing the predictor's output space and iteratively adjusting its output values. However, this discretization approach departs from the standard empirical risk minimization (ERM) pipeline, introduces rounding error and an additional sensitive hyperparameter, and may distort the predictor's outputs in ways that hinder downstream decision-making.

In this work, we propose a discretization-free multicalibration method that directly optimizes an empirical risk objective over an ensemble of depth-two decision trees. Our ERM approach can be implemented using off-the-shelf tree ensemble learning methods such as LightGBM. Our algorithm provably achieves multicalibration, provided that the data distribution satisfies a technical condition we term as loss saturation. Across multiple datasets, our empirical evaluation shows that this condition is always met in practice. Our discretization-free algorithm consistently matches or outperforms existing multicalibration approaches—even when evaluated using a discretization-based multicalibration metric that shares its discretization granularity with the baselines. Code to replicate the results in this work is available at `https://github.com/hjenryin/Discretization-free-MC`.

## 1 Introduction

In many applications, machine learning predictors are at the heart of the decision-making process, from finance (Fuster et al., 2022) to healthcare diagnostics (Rajkomar et al., 2018). Despite its ubiquity, there is growing concern that these predictors might discriminate against individuals in protected groups. In recent years, *multicalibration* (Hebert-Johnson et al., 2018) has emerged from the algorithmic fairness literature as a learning objective to mitigate the risk of algorithmic discrimination. Informally, multicalibration requires a predictor to be calibrated on average over a family of groups $\mathcal{G}$: for all groups $g \in \mathcal{G}$ and for all $v$ in the range of $f$,

$$\mathbb{E}[(y - f(x)) \cdot g(x)|f(x) = v] = 0$$

where $g \colon \mathcal{X} \to \{0, 1\}$ is a group indicator function.

---

[*]Part of the work was done at CMU.

39th Conference on Neural Information Processing Systems (NeurIPS 2025).

Typically, multicalibration algorithms discretize the output space of the predictor so that the range of $f$, $R(f)$, is a finite set. Existing work in the multicalibration literature (Hebert-Johnson et al., 2018; Globus-Harris et al., 2023; Haghtalab et al., 2023) heavily relies on the discretized output space or *level sets*, and all existing algorithms use calibration data to iteratively remap the output of a predictor $f$ within these level sets to reduce the multicalibration error at each iteration. However, discretization can be undesirable in practice for several reasons. First, it introduces rounding error that can distort predictions and degrade accuracy. Second, it adds a sensitive hyperparameter—the discretization granularity—that must be carefully tuned. As noted by Nixon et al. (2019), this induces a bias-variance tradeoff: finer discretization improves resolution but requires more data per bin, while coarser discretization sacrifices precision. Błasiok & Nakkiran (2023) observe that the calibration error metric can be unstable under different discretization parameters. Finally, when there are multiple downstream decision makers with heterogeneous utility functions, fixed discretization may fail to provide the precision necessary for them to make optimal decisions. (Zhao et al., 2021)

**Our results and contributions.** In a nutshell, we develop a simple, practical, and performant discretization-free algorithm for multicalibration. In more details:

- Given a black-box predictor, our discretization-free algorithm post-processes its outputs by solving a square-loss empirical risk minimization (ERM) problem over an ensemble of depth-two decision trees. Each tree splits on features derived from the predictor's output or group membership. This ERM step can be efficiently implemented using standard tree ensemble methods such as LightGBM (Ke et al., 2017).

- We prove theoretically that our algorithm outputs a multicalibrated predictor, given the assumption that decision tree ensembles saturate in loss improvement after optimization–that is, the square loss cannot be further reduced through another round of tree-ensemble post-processing. Empirically, we observe that this saturation condition consistently holds across multiple real-world datasets.

- We evaluate our discretization-free approach across a diverse set of tabular, image, and text datasets. Compared to existing multicalibration methods, our algorithm delivers competitive and often lower multicalibration error—even when baselines are tuned using the same discretization scheme used for evaluation.

## 2 Related works

**Multicalibration** Multicalibration was first introduced by Hebert-Johnson et al. (2018) as a notion of multi-group fairness. This growing line of research is then extended in several directions, including generalizing or relaxing the notion of multicalibration (Gopalan et al., 2022; Zhang et al., 2024; Wu et al., 2024; Deng et al., 2023), applying multicalibration to conformal prediction (Jung et al., 2022), as well as exploring mathematical implications of multicalibration (Jung et al., 2022; Gopalan et al., 2021; Globus-Harris et al., 2023; Wu et al., 2024; Kim et al., 2022).

**Algorithms for multicalibration.** All existing multicalibration algorithms are discretization-based and operate across level sets of the predictor. Hebert-Johnson et al. (2018); Globus-Harris et al. (2023) audit multicalibration error of the predictor within the level sets in each iteration and patch identified bias until convergence. Haghtalab et al. (2023) views multicalibration as a multi-objective optimization problem solved using game dynamics, where the groups and the level sets are generalized to distributions and loss functions in general.

**Multicalibration and Loss Minimization.** A key appeal of multicalibration is its alignment with loss minimization: existing multicalibration algorithms iteratively update the predictor with each update reducing the square loss. This naturally raises the question—can multicalibration be achieved directly via a single loss minimization step? Błasiok et al. (2023) show that minimizing population loss over a class of neural networks of certain size yields multicalibration for groups defined by smaller networks. However, their result is non-constructive and assumes optimization oracle over an intractable function class. In contrast, our approach solves a one-shot ERM over a simple class of tree ensembles, implementable via standard tools. Relatedly, Hansen et al. (2024) ask whether ERM alone suffices. Our experiments show that it often does not—but our post-processing reliably recovers multicalibration.

## 3 Preliminaries

We consider prediction tasks over a domain $\mathcal{Z} = \mathcal{X} \times \mathcal{Y}$, where $\mathcal{X}$ represents the feature domain and $\mathcal{Y}$ represents the label domain. We require the labels to be real values in $[0, 1]$ – that is, the label can either be a binary outcome for classification with $\mathcal{Y} = \{0, 1\}$ or a real value for regression with $\mathcal{Y} = [0, 1]$. For a domain $\mathcal{Z}$, we write $\mathcal{D} \in \Delta(\mathcal{Z})$ to denote the true distribution over the labeled samples.

A predictor $f$ is a map $f : \mathcal{X} \to [0, 1]$. We write $\mathcal{R}(f)$ to denote the range of a predictor $f$. We are interested in the squared error of a predictor $f$ with respect to the underlying distribution $\mathcal{D}$.

**Definition 3.1** (Squared loss). *The squared loss of a predictor $f$ on distribution $\mathcal{D}$ is*

$$\ell(f, \mathcal{D}) = \mathbb{E}_{(x,y)\sim\mathcal{D}}[(f(x) - y)^2] \tag{1}$$

*We omit the distributions $\mathcal{D}$ from the notation when it is clear from context.*

Note that when $\mathcal{Y} = [0, 1]$, the squared loss is the familiar MSE loss function, whose application in multicalibration has been discussed in Globus-Harris et al. (2023). When $\mathcal{Y} = \{0, 1\}$, the squared loss is the Brier score, which is commonly used for multicalibration algorithms (Hansen et al., 2024; Błasiok et al., 2023).

We formally specify the multicalibration notion used in this work. Given the underlying distribution $\mathcal{D}$ and a set of groups $\mathcal{G} \subset 2^{\mathcal{X}}$, we write $\boldsymbol{g} : \mathcal{X} \to \{0, 1\}^{|G|}$ as the high-dimensional group indicator function, where $g_i(x) = 1$ indicates that the datapoint $x \in \mathcal{X}$ belongs to the $i$-th group. To measure the multicalibration error, we use the following $\ell_1$ definition (Hansen et al., 2024). Note that the $\ell_1$ version of multicalibration is more interpretable than the $\ell_2$ version, and different versions of multicalibration can be bounded by each other (Globus-Harris et al., 2023).

**Definition 3.2** (Multicalibration Error). *Fix a distribution $\mathcal{D} \in \Delta(\mathcal{Z})$ and a predictor $f : \mathcal{X} \to [0, 1]$. We define the multicalibration error of $f$ with respect to $\mathcal{D}$ and $\mathcal{G}$ as:*

$$\max_{i\in[|G|]} \sum_{v\in\mathcal{R}(f)} \Pr_{(x,y)\sim\mathcal{D}}[f(x) = v, g_i(x) = 1] \cdot \left|\mathbb{E}_{(x,y)\sim\mathcal{D}}\left[f(x) - y \mid f(x) = v, g_i(x) = 1\right]\right| \tag{2}$$

We say that a predictor $f$ is $\alpha$-multicalibrated if its multicalibration error is at most $\alpha$. Note that our definitions here and analysis in the next section are population-based, and we leave the finite-sample analysis to Appendix F.

## 4 DFMC: A Discretization-Free Algorithm for Multicalibration

In this section, we describe our proposed algorithm for multicalibration. At a high level, we train a decision tree ensemble of depth 2, using the output of a base predictor $f_0$ and the group membership as input features, to minimize the squared loss $\ell$ on the true distribution $\mathcal{D}$.

More formally, consider the following subsets of $\mathcal{X}$ used as the splitting criterion for the decision trees:

$$\begin{aligned}\mathcal{S}_1(f_0) &= \left\{\{x \in \mathcal{X} : f_0(x) \geq v\} : v \in \mathcal{R}(f_0) \cup \{0\}\right\}, \\ \mathcal{S}_2(\mathcal{G}) &= \left\{\{x \in \mathcal{X} : g_i(x) = 1\} : i \in [|G|]\right\},\end{aligned} \tag{3}$$

Each decision tree of depth 2 in the ensemble assigns a real value $c_i : i \in [4]$ to each of its 4 leaves. Formally, the set of decision trees is written as $\mathcal{T}(f_0, \mathcal{G})$, where:

$$\begin{aligned}\mathcal{T}(f_0, \mathcal{G}) = \Big\{&c_1 \cdot \mathbb{I}\{x \in s_1 \wedge x \in s_2\} + c_2 \cdot \mathbb{I}\{x \in s_1 \wedge x \notin s_2\} + c_3 \cdot \mathbb{I}\{x \notin s_1 \wedge x \in s_2\} \\ &+ c_4 \cdot \mathbb{I}\{x \notin s_1 \wedge x \notin s_2\} : c_1, c_2, c_3, c_4 \in \mathbb{R}, s_1 \in \mathcal{S}_1(f_0), s_2 \in \mathcal{S}_2(\mathcal{G})\Big\}\end{aligned} \tag{4}$$

To find the optimal set of decision trees $T \subseteq \mathcal{T}(f, \mathcal{G})$, we use a solver to minimizes the squared loss and obtain our post-processed predictor:

$$p_{\mathcal{G}}(f_0) = \operatorname*{arg\,min}_{T\subseteq\mathcal{T}(f_0,\mathcal{G})} \ell\left(f_0 + \sum_{t\in T} t, \mathcal{D}\right) \tag{5}$$

In practice, the implementation of most solvers does not enforce this exact structure of choosing splitting criterion. For simplicity, we present our algorithm as in Equation (4), and we explain in Appendix C why our analysis still holds for them despite such slight differences.

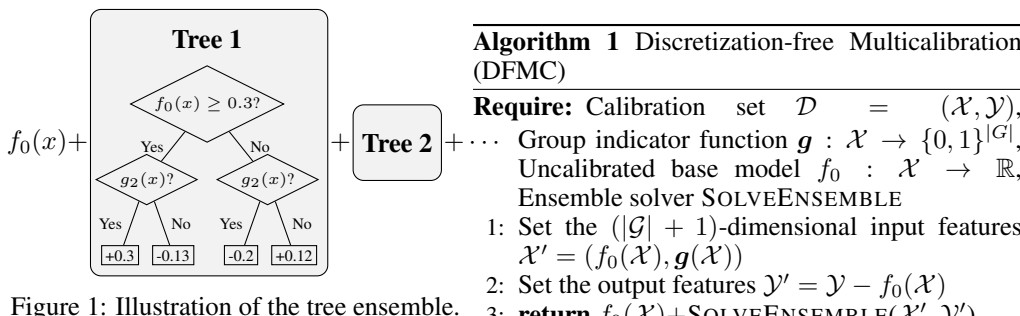

Figure 1: Illustration of the tree ensemble.

**Algorithm 1** Discretization-free Multicalibration (DFMC)

**Require:** Calibration set $\mathcal{D} = (\mathcal{X}, \mathcal{Y})$, Group indicator function $g : \mathcal{X} \to \{0,1\}^{|G|}$, Uncalibrated base model $f_0 : \mathcal{X} \to \mathbb{R}$, Ensemble solver SOLVEENSEMBLE
1: Set the $(|\mathcal{G}| + 1)$-dimensional input features $\mathcal{X}' = (f_0(\mathcal{X}), g(\mathcal{X}))$
2: Set the output features $\mathcal{Y}' = \mathcal{Y} - f_0(\mathcal{X})$
3: **return** $f_0(\mathcal{X}) + $ SOLVEENSEMBLE$(\mathcal{X}', \mathcal{Y}')$

An illustration of Equation (5) can be found in Figure 1. Being discretization-free enables our algorithm to avoid the complexity with discretization, makes it simpler and provides more flexibility than existing ones. Algorithm 1 outlines how we can optimize for Equation (5) using on-the-shelf solvers like LightGBM (Ke et al., 2017) or XGBoost (Chen & Guestrin, 2016) with simple features. Also, although the evaluation of Definition 3.2 still requires discretized predictions, our discretization-free predictor can be evaluated under flexible discretization scheme with consistently strong performance. We formalize this flexibility in Definition 4.1.

**Definition 4.1** (Discretization Operation). *A function $\tilde{f}_m$ is an $m$-discretized version of a predictor $f$ if it maps the continuous output of $f$ to a finite set of $m$ values, with the property that $\tilde{f}_m(x_1) \geq \tilde{f}_m(x_2)$ whenever $f(x_1) \geq f(x_2)$ for any $x_1, x_2 \in \mathcal{X}$. We write the discretization error $\epsilon_{round}$ as*

$$\epsilon_{round} = \ell(\tilde{f}_m^{cal}, \mathcal{D}) - \ell(f^{cal}, \mathcal{D}), \tag{6}$$

Note that we do not restrict ourselves to a specific discretization function for this evaluation. Rounding to the nearest value on a predefined grid, which is commonly used in existing multicalibration algorithms, is among the possible discretization methods we allow. As we will show shortly, unlike algorithms that incorporate discretization as an intrinsic component, our discretization-free predictor performs well under various discretization schemes used solely for evaluation purposes.

Through the following lemmas, we associate the continuous optimization for the squared loss with the discretized metric of multicalibration.

**Lemma 4.2.** *Consider the post-processing step $p_{\mathcal{G}}$ in Equation* (5). *For all $f \in [0,1]$ and $m \in \mathbb{N}$, and for any $m$-discretized version $\tilde{f}_m$ of $f$, we have*

$$\ell(p_{\mathcal{G}}(f), \mathcal{D}) \leq \ell(p_{\mathcal{G}}(\tilde{f}_m), \mathcal{D})$$

The proof follows by constructing a set of trees $T \in \mathcal{T}(f, \mathcal{G})$ such that $f + \sum_{t \in T} t$ approximates $p_{\mathcal{G}}(\tilde{f}_m)$ well.

**Lemma 4.3.** *Given a predictor $f$ whose range is of size $m$, training an ensemble of decision trees on $f$ as in the post-processing step in Equation* (5) *encompasses adding a linear function of the groups on each output level set of $f$. That is,*

$$\left\{ f + \sum_{j=1}^{m} \mathbb{I}\{f(x) = v_j\} \left( c^j + \sum_i c_i^j g_i \right) \ \middle| \ c_i^j \in \mathbb{R} \right\} \subseteq \left\{ f + \sum_{t \in T} t \ \middle| \ T \subseteq \mathcal{T}(f, \mathcal{G}) \right\} \tag{7}$$

*where $v_1, \cdots v_m \in \mathcal{R}(f), v_1 < v_2 < \cdots < v_m$.*

The proof follows by rewriting $\mathbb{I}\{f(x) = v_j\}$ as thresholding functions.

**Lemma 4.4.** *If a predictor $f$ with $\mathcal{R}(f) = \{v_1, \cdots, v_m\}$ has a multicalibration error w.r.t. $\mathcal{G}$ larger than $\alpha$, then there exists linear functions $h_j = \sum_i c_i^j g_i$, such that*

$$\ell(f, \mathcal{D}) > \ell\left(f + \sum_{j=1}^m \mathbb{I}\{f = v_j\} h_j, \mathcal{D}\right) + \alpha^2. \tag{8}$$

The proof follows by picking the worst group $g_k$ from the definition of multicalibration for each $v_j$ and constructing the linear functions $h_j = c_k^j g_k$.

Before the main theorem, we introduce the following necessary assumption.

**Assumption 4.5** (Loss Saturation of Algorithm 1). *Given an uncalibrated predictor $f_0$, let $f^{cal} = p_{\mathcal{G}}(f_0)$ be our proposed predictor calibrated with Algorithm 1. We assume that the squared loss of $f^{cal}$ is $\epsilon$-saturated w.r.t $p_{\mathcal{G}}$ with a small marginal improvement $\epsilon_{loss} \ll 1$:*

$$\ell(f^{cal}, \mathcal{D}) \leq \ell(p_{\mathcal{G}}(f^{cal}), \mathcal{D}) + \epsilon_{loss} \tag{9}$$

*that is, running Algorithm 1 on $f^{cal}$ again gives a small marginal improvement of at most $\epsilon_{loss}$.*

This is a reasonable assumption because our formulation is essentially a supervised learning problem with simple features, where the objective is to minimize the squared loss. Decision tree ensembles are a well-established and widely used method in general for this setting, with mature solvers and have stood the test of time. Consequently, the loss achieved by our method should already be low, leaving little room for further improvement. We also provide a numerical simulation in Table 1 on the datasets to empirically support this assumption.

**Theorem 4.6** (Algorithm 1 Yields Small Multicalibration Error Given Assumption 4.5). *Let $f^{cal}$ be the predictor obtained by Algorithm 1, and $\tilde{f}_m^{cal}$ be its $m$-discretized version. If the discretization error of $\tilde{f}_m^{cal}$ is $\epsilon_{round}$, and the loss of $f^{cal}$ is $\epsilon_{loss}$-saturated with respect to $p_{\mathcal{G}}$ as in Assumption 4.5, then the multicalibration error of $\tilde{f}_m^{cal}$ with respect to $\mathcal{G}$ is at most $\sqrt{\epsilon_{loss} + \epsilon_{round}}$.*

*Proof.* If $\tilde{f}_m^{cal}$ has a multicalibration error larger than $\alpha$, then post-processing it with $p_{\mathcal{G}}$ can reduce the loss:

$$
\begin{aligned}
\ell(\tilde{f}_m^{cal}, \mathcal{D}) - \ell\left(p_{\mathcal{G}}(\tilde{f}_m^{cal}), \mathcal{D}\right) =& \ell(\tilde{f}_m^{cal}, \mathcal{D}) - \min_{T \subseteq \mathcal{T}(\tilde{f}_m^{cal}, \mathcal{G})} \ell\left(\tilde{f}_m^{cal} + \sum_{t \in T} t, \mathcal{D}\right) && \text{(Equation (5))} \\
\geq& \ell(\tilde{f}_m^{cal}, \mathcal{D}) - \min_{h_j} \ell\left(\tilde{f}_m^{cal} + \sum_{j=1}^m \mathbb{I}\{\tilde{f}_m^{cal} = v_j\} h_j, \mathcal{D}\right) && \\
&&& \text{(Lemma 4.3)} \\
>& \alpha^2 && \text{(Lemma 4.4)}
\end{aligned}
$$

On the other hand, the reduction of loss should be small:

$$
\begin{aligned}
\ell(\tilde{f}_m^{cal}, \mathcal{D}) - \ell\left(p_{\mathcal{G}}(\tilde{f}_m^{cal}), \mathcal{D}\right) \leq& \ell(f^{cal}, \mathcal{D}) + \epsilon_{round} - \ell\left(p_{\mathcal{G}}(\tilde{f}_m^{cal}), \mathcal{D}\right) && \text{(Equation (6))} \\
\leq& \ell(f^{cal}, \mathcal{D}) + \epsilon_{round} - \ell\left(p_{\mathcal{G}}(f^{cal}), \mathcal{D}\right) && \text{(Lemma 4.2)} \\
\leq& \epsilon_{loss} + \epsilon_{round} && \text{(Assumption 4.5)}
\end{aligned}
$$

This implies that the multicalibration error of $\tilde{f}_m^{cal}$ is at most $\sqrt{\epsilon_{loss} + \epsilon_{round}}$. $\qquad\square$

**Remark 4.7.** *To wrap up the analysis, we provide a high level comparison of our algorithm with relevant prior works. The theoretical algorithm proposed in Błasiok et al. (2023) is the most similar one to ours in that both theoretically achieve discretization-free multicalibration through loss minimization. While they theoretically proved loss saturation for an intractable class of neural networks by adding more nodes without practical implementation, we empirically verify this assumption for tree ensembles and provide an efficient algorithm.*

*On the other hand, the construction of a tree ensemble is highly relevant to the algorithm presented in Globus-Harris et al. (2023). Unlike their approach, ours uses the continuous output as a feature in the ensemble, thereby avoiding discretization. Apart from that, our algorithm is quite similar to the first iteration of theirs via the relationship shown in Lemma 4.3, with Assumption 4.5 indicating convergence after this single iteration.*

# 5   Experiments

## 5.1   Dataset Setup

In this section, we describe the dataset we use and the experiment pipeline.

A common dataset to consider in fairness and multicalibration literature is the ACS dataset, obtained through the Folktables (Ding et al., 2021) package. The Folktables package defines a rich set of tasks, and we consider three tasks derived from the dataset: **income regression** and **travel time regression** predicts the person's income and commute time, and **income classification** predicts whether a person's income is higher than $50,000. On each of these tasks, we define around 50 groups based on the data attributes.

The prior work of Hansen et al. (2024) has pointed out that multicalibration post-processing has limited effect on tabular data, so we also evaluate our algorithm on various text and image datasets. Zhang et al. (2017) introduced the **UTKFace** dataset, with a person's face image associated with their age, gender and race. We consider the task of predicting the age of the person given the face, and use individual's attribute as the group. ISIC challenge dataset is an image dataset related to skin lesion, and we considered the task introduced in the **ISIC Challenge 2019** (Tschandl et al., 2018; Codella et al., 2017; Combalia et al., 2019) to predict whether the skin lesion is a melanocytic nevus or not. The dataset provide metadata including the age, gender or anatomic site of the patient, and we use these as the group. Finally, we consider the **Comment Toxicity Classification** dataset (Borkan et al., 2019) from the WILDS dataset (Koh et al., 2021), which is a text classification task to predict whether a comment is toxic or not. The WILDS dataset already defines eight groups based on the identities mentioned in the comment, and we use them as the group.

Detailed description of the dataset, including dataset partitioning and the specific groups, is left to Appendix D.

## 5.2   Our algorithms and the baselines

In this section we describe the algorithms evaluated in these experiments, which includes two discretization-free baselines with no multicalibration guarantee, two existing multicalibration algorithms, and our algorithm. The **uncalibrated baseline** depends largely on the task: We used linear models for tabular data, ResNet for image data, and DistilBERT for text data. Due to its dependency on the specific dataset, further details are left to Appendix D. Another baseline algorithm that we might be interested in is the **multiaccurate predictor** (Hebert-Johnson et al., 2018; Roth, 2022), which guarantees that the predicted mean on each subgroup is close to the true mean on that group. Although it's a far weaker notion than multicalibration, it's an extremely simple method promoting fairness across subgroups that can be implemented with a single linear model.

The multicalibration algorithms that we compare with are MCBoost and LSBoost. Both require discretizing the output space in advance and work in an iterative manner by correcting the prediction on each level set until convergence. **MCBoost** (Roth, 2022) enumerates over $\{x : g_i(x) = 1, f(x) = v\}$ for all $i \in [|\mathcal{G}|]$ and $v \in \mathcal{R}(f)$ in every single iteration and fixes the subset with the largest deviation from the true mean by setting the predicted value on that subset to the true mean. **LSBoost** (Globus-Harris et al., 2023), on the other hand, considers the groups as a whole. In each iteration, the whole calibration set is partitioned into multiple level sets based on the current predicted value, and a weak learner is fit with the group information for each level set using data points in that set. The predicted value of the weak learner will assign the data points to a new level set for the next iteration. Finally, we implement **our algorithm** described in Algorithm 1 with the LightGBM (Ke et al., 2017) solver.

MCBoost and LSBoost require discretization before calibration, so for each selection of discretization, we find the hyperparameters that minimize the multicalibration error on the validation set. As

Table 1: The empirical marginal improvement of running Algorithm 1 the second time. The improvement is minimal, validating Assumption 4.5.

| Task | $\ell(f_0)$ | $\ell(f^{cal})$ | $\ell(p_{\mathcal{G}}(f^{cal}))$ | $\hat{\epsilon}_{loss} = \ell(f^{cal}) - \ell(p_{\mathcal{G}}(f^{cal}))$ |
|---|---|---|---|---|
| Skin Lesion Classification | 0.2262 | 0.1475 | 0.1492 | $-2 \times 10^{-3}$ |
| Income Classification | 0.1831 | 0.1432 | 0.1431 | $1 \times 10^{-4}$ |
| Comment Toxicity Classification | 0.0589 | 0.0516 | 0.0517 | $-2 \times 10^{-5}$ |
| Age Regression | 0.0051 | 0.0011 | 0.0011 | $7 \times 10^{-6}$ |
| Income Regression | 0.0414 | 0.0361 | 0.0361 | $6 \times 10^{-7}$ |
| Travel Time Regression | 0.0302 | 0.0298 | 0.0298 | $2 \times 10^{-6}$ |

for the other three algorithms, we just directly minimize the squared error. Further details, including training specifications and hyperparameters, are left to Appendix E. For all experiments, we fix the training set for the uncalibrated baseline and the test set for evaluation. The calibration and validation set are partitioned 10 times, with the standard deviation indicated in the plots and the tables.

## 5.3 Experiment Results

### 5.3.1 Empirical validation of Assumption 4.5

We first validate Assumption 4.5 empirically: Running Algorithm 1 a second time gives small marginal improvement. Note that due to the imperfection of optimization, we would not be able to obtain the true minimizer of the test set. Instead, we run the optimization on the calibration set with early stopping and present the empirical loss improvement $\hat{\epsilon}_{loss} = \ell(\hat{p}_{\mathcal{G}}(f)) - \ell(f)$ on the test set. The true $\epsilon_{loss}$ would contain an additional optimization error term $\epsilon_{opt}$, satisfying $\epsilon_{loss} = \hat{\epsilon}_{loss} + \epsilon_{opt}$. We present $\hat{\epsilon}_{loss}$ in Table 1 and summarize our first observation:

**Observation 1.** *Across all different datasets and tasks evaluated, the marginal improvement of running Equation* (5) *a second time is small, validating the loss saturation assumption in Assumption 4.5.*

### 5.3.2 Threshold-robust evaluation of multicalibration error

In this section, we empirically validate Theorem 4.6 and compared our discretization-free algorithm with other multicalibration algorithms. Recall that for the evaluation of multicalibration error, the predictor needs to have a finite range, so we would need to derive a discretized version of a continuous predictor for evaluation. We ensure a fair comparison across different algorithms by using the same discretization regime. In particular, we use the grid discretization regime and restrict the codomain to $LS = \{\frac{1}{2m}, \frac{3}{2m}, \cdots, \frac{2m-1}{2m}\}$, where $m$ is the number of element in the level set. We vary $m \in \{10, 20, 30, 50, 75, 100\}$. While a separate predictor is trained for each $m$ for MCBoost and LSBoost, we only train a single predictor for our algorithm and then evaluate the multicalibration error of its $m$-discretized version with varying $m$.

We present the multicalibration error versus the number of non-empty level sets in Figure 2.

**Observation 2.** *Although empirical baseline algorithms can obtain a low multicalibration error in certain cases, post-processing can further decrease the error in general.*

A direct observation is that *Travel Time Regression* is an outlier among the six subplots in Figure 2. Although in all other datasets evaluated, the uncalibrated baseline has a higher multicalibration error, compared to other algorithms, such a difference in the error is not observed in this dataset. The uncalibrated baseline algorithm already obtained low multicalibration error, leaving little room for improvement for other post-processing methods. This partially echoes the observations in Hansen et al. (2024) that ERM on certain function classes can already achieve a low multicalibration error on tabular data; yet it also highlights that post-processing is indeed necessary in general to obtain a multicalibrated predictor.

**Observation 3.** *Our algorithm achieves multicalibration error that matches and often improves upon existing methods. This performance advantage persists even when competing algorithms are specifically tuned using the same discretization scheme used in evaluation.*

## Multicalibration error

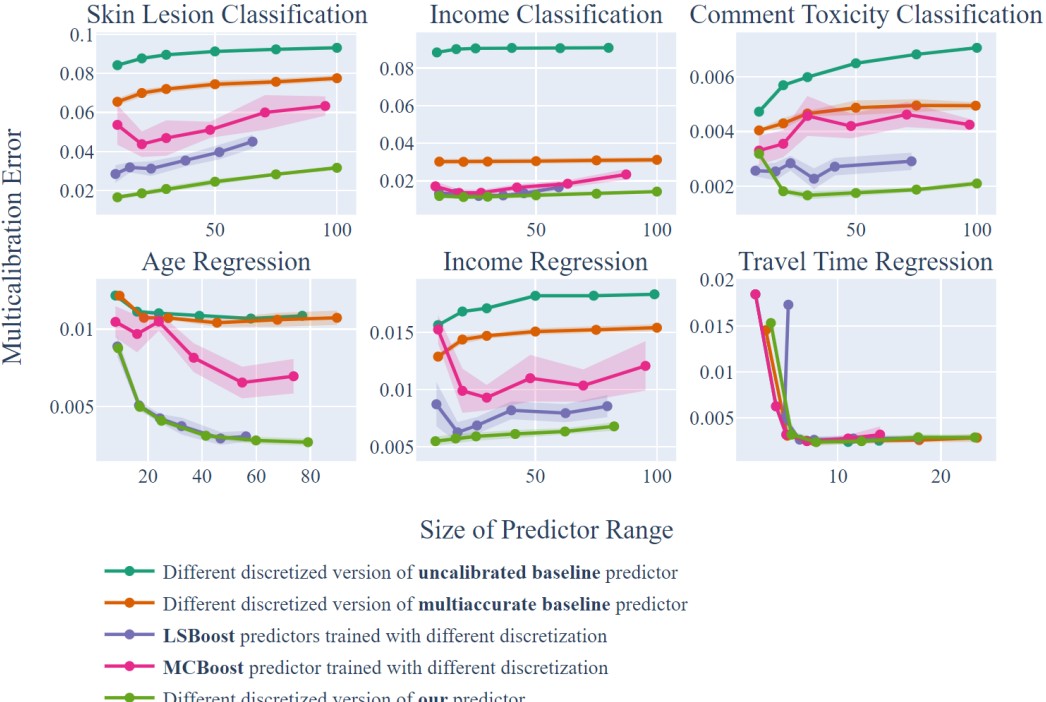

Figure 2: Multicalibration error on different datasets. The y-axis is the multicalibration error, and the x-axis is the size of range, which may not equal $m$, the size of the codomain, and the error band displays the standard deviation. The graph shows that, a predictor trained using our algorithm can be discretized arbitrarily, and its $m$-discretized version has matching or lower multicalibration error than the predictor calibrated with other multicalibration algorithm using that specific discretization.

We compare the algorithms at the same range size to minimize the effect of discretization error. Our discretization-free algorithm was only calibrated once to minimize the squared loss, and in most cases its $m$-discretized version has matching or better multicalibration error than the other multicalibration algorithms, even though these algorithms go through a separate hyperparameter tuning and calibration process for each discretization $m$. This observation validates our theoretical analysis in Theorem 4.6 and demonstrates the benefit of our discretization-free algorithm of being flexible to down-stream discretization.

### 5.3.3 Discretization-free fairness through worst-group Smooth ECE

We would also like to evaluate our algorithm with another metric, **worst-group smooth expected calibration error (worst-group smECE)**, which has also been used as a heuristic metric in Hansen et al. (2024). smECE was introduced by Błasiok & Nakkiran (2023) as a continuous metric of calibration, and thus we feel it can be suitable for our evaluation. In the context of multi-group fairness, we could evaluate smECE on all groups that we're interested in and take the worst group as the final metric, although it is not directly related to multicalibration. This would also serve as a complement to the metric of multicalibration, which can be viewed as the group-size-weighted worst-group binned ECE. The result presented in Table 2 can be summarized by the following observation.

**Observation 4.** *Our discretization-free algorithm achieves better calibration on the worst groups than other multicalibration algorithms.*

Table 2: The worst-group smECE of the evaluated algorithms, along with their standard deviation. The values have been multiplied by 1000 for readability. Results show that our discretization-free algorithm is more calibrated among the worst groups than other multicalibration algorithms.

| Method | Discretization | Skin Lesion Class. | Income Class. | Comment Toxicity Class. | Age Reg. | Income Reg. | Travel Time Reg. |
|---|---|---|---|---|---|---|---|
| Uncalibrated Baseline | / | 290.77 | 281.81 | 119.28 | 111.04 | 155.13 | 72.38 |
| Multiaccurate Baseline | / | 169.28 ±6.24 | 94.05 ±5.71 | 71.38 ±6.89 | 47.54 ±0.71 | 56.61 ±0.29 | 34.94 ±5.72 |
| LSBoost | 10 | 146.68 ±27.44 | 102.06 ±11.17 | 41.04 ±10.34 | 41.51 ±9.73 | 72.29 ±13.41 | 58.73 ±3.59 |
| | 20 | 159.31 ±29.24 | 102.57 ±17.25 | 39.39 ±12.74 | 23.29 ±7.00 | 73.82 ±9.76 | 54.96 ±14.69 |
| | 30 | 160.80 ±36.54 | 90.17 ±6.59 | 36.61 ±9.77 | **20.25** ±4.71 | 80.03 ±13.36 | 60.88 ±13.63 |
| | 50 | 147.87 ±30.32 | 102.08 ±12.13 | 38.42 ±5.12 | 25.49 ±8.76 | 82.78 ±7.35 | 55.20 ±11.82 |
| | 75 | 144.05 ±22.57 | 108.26 ±14.03 | 45.66 ±6.55 | 34.16 ±7.26 | 81.47 ±3.62 | 59.07 ±8.73 |
| | 100 | 149.77 ±26.00 | 124.94 ±20.48 | 44.28 ±7.77 | 51.13 ±11.13 | 76.83 ±3.39 | 66.59 ±5.34 |
| MCBoost | 10 | 278.75 ±17.81 | 259.83 ±5.00 | 78.30 ±15.59 | 106.73 ±0.00 | 154.28 ±5.12 | 59.42 ±0.00 |
| | 20 | 263.71 ±18.93 | 212.19 ±33.65 | 65.17 ±20.58 | 108.20 ±0.00 | 138.96 ±6.81 | 72.01 ±0.00 |
| | 30 | 261.02 ±26.77 | 186.57 ±41.96 | 90.66 ±17.87 | 106.34 ±0.00 | 125.01 ±16.07 | 73.96 ±0.00 |
| | 50 | 252.76 ±17.20 | 182.07 ±45.79 | 86.74 ±21.29 | 112.37 ±0.00 | 141.07 ±5.60 | 69.69 ±6.83 |
| | 75 | 239.26 ±22.52 | 193.78 ±47.80 | 81.74 ±6.30 | 111.00 ±3.72 | 130.15 ±16.26 | 68.79 ±2.25 |
| | 100 | 259.49 ±21.55 | 198.78 ±31.04 | 85.24 ±9.86 | 109.82 ±3.79 | 131.43 ±11.99 | 67.30 ±10.89 |
| Ours | / | **89.26** ±15.63 | **87.30** ±25.94 | **24.57** ±4.76 | 20.87 ±3.14 | **44.41** ±5.44 | **31.85** ±7.26 |

### 5.3.4 Evaluation of loss and accuracy

Apart from multicalibration, some standard metrics might be of interest when evaluating an algorithm. We present accuracy for classification tasks and MSE error for regression tasks. The results presented in Table 3 yields the following observation:

**Observation 5.** *Our algorithm does not sacrifice standard evaluation metrics like MSE error or accuracy.*

Such results are anticipated, as our algorithm is simply minimizing the loss (namely the MSE error for regression tasks or the Brier score for classification tasks) based on a set of selected features.

## 6 Conclusion

In this work, we provide a discretization-free algorithm for multicalibration by solving an ERM problem with simple features using decision tree ensembles, thereby avoiding the nuisance of discretization. We prove theoretically that our algorithm outputs a multicalibrated predictor, given the assumption of loss saturation in tree ensembles, which is consistently validated across multiple real-world datasets. Our experiments also confirm our algorithm's strong performance compared to existing ones.

We believe our work has impact on societal aspects of machine learning. Our algorithm can be applied to high-stakes decision-making systems in domains such as finance, healthcare, and employment. The discretization-free nature allows our predictor to serve as a drop-in replacement for an uncalibrated regressor or binary classifier suitable for different downstream applications; and by framing multicalibration into a typical loss minimization problem, we provide a simple library-calling

Table 3: The accuracy (%) of the evaluated algorithms, along with their standard deviation. Results show that our discretization-free algorithm does not interfere with the standard MSE error of regression problems.

| Method | Discretization | Class. Accuracy (%) | | | Reg. MSE Error ($\times 10^{-3}$) | | |
|---|---|---|---|---|---|---|---|
| | | Skin Lesion Class. | Income Class. | Comment Toxicity Class. | Age Reg. | Income Reg. | Travel Time Reg. |
| Uncalibrated Baseline | / | 70.21 | 76.71 | 92.88 | 5.11 | 41.39 | 30.19 |
| Multiaccurate Baseline | / | 73.19 ±0.31 | 78.97 ±0.06 | 92.89 ±0.00 | 3.51 ±0.02 | 37.92 ±0.02 | **29.76** ±0.02 |
| LSBoost | 10 | 77.22 ±1.00 | 77.96 ±0.25 | 92.81 ±0.08 | 2.01 ±0.24 | 39.66 ±0.35 | 31.06 ±0.07 |
| | 20 | 76.55 ±0.95 | 78.09 ±0.33 | 92.78 ±0.06 | 1.42 ±0.16 | 38.98 ±0.31 | 30.29 ±0.09 |
| | 30 | 76.86 ±0.46 | 78.16 ±0.21 | 92.78 ±0.06 | 1.36 ±0.13 | 39.03 ±0.23 | 30.25 ±0.04 |
| | 50 | 77.06 ±0.52 | 78.11 ±0.20 | 92.71 ±0.07 | 1.52 ±0.17 | 39.51 ±0.22 | 30.16 ±0.03 |
| | 75 | 76.52 ±0.72 | 77.81 ±0.27 | 92.65 ±0.04 | 1.53 ±0.19 | 38.83 ±0.19 | 30.13 ±0.03 |
| | 100 | 76.12 ±0.50 | 77.60 ±0.30 | 92.81 ±0.05 | 1.72 ±0.20 | 38.80 ±0.20 | 30.17 ±0.03 |
| MCBoost | 10 | 71.00 ±1.01 | 76.71 ±0.00 | 92.88 ±0.00 | 5.08 ±0.25 | 42.18 ±0.01 | 31.23 ±0.00 |
| | 20 | 72.15 ±1.63 | 77.10 ±0.34 | 92.88 ±0.00 | 4.69 ±0.30 | 41.10 ±0.23 | 30.42 ±0.00 |
| | 30 | 73.54 ±1.24 | 77.23 ±0.20 | 92.88 ±0.00 | 4.92 ±0.17 | 40.70 ±0.26 | 30.31 ±0.00 |
| | 50 | 72.07 ±1.01 | 77.20 ±0.40 | 92.88 ±0.00 | 4.40 ±0.38 | 40.93 ±0.27 | 30.22 ±0.00 |
| | 75 | 73.12 ±1.00 | 76.89 ±0.17 | 92.88 ±0.00 | 3.81 ±0.36 | 40.72 ±0.32 | 30.22 ±0.01 |
| | 100 | 72.29 ±1.08 | 76.70 ±0.22 | 92.87 ±0.01 | 3.88 ±0.41 | 40.95 ±0.29 | 30.18 ±0.03 |
| Ours | / | **78.90** ±0.17 | **79.27** ±0.07 | **92.93** ±0.01 | **1.08** ±0.02 | **36.11** ±0.11 | 29.80 ±0.04 |

implementation that lowers the technical barrier for practitioners to incorporate fairness-aware techniques into their workflows. We believe this work can enhance the adoption of multicalibration techniques within the machine learning community.

A limitation of this work is that Assumption 4.5 is a heuristic assumption that holds under realistic settings, but has unrealistic counterexamples described in Appendix B where $f_0$ contains no information. From this construction, it seems that the additional improvement $\epsilon_{loss}$ may depend on the uncalibrated predictor we start with, namely $f_0$. For future work, it would be interesting to formalize such dependency and bound the error.

Additionally, the scope of this work primarily focuses on group fairness, requiring binary group indicators to be explicitly provided as the input to the proposed algorithm, which is a common setting in group fairness research. (Caton & Haas, 2024) The algorithm does have the potential to extend to implicit groups, such as groups defined from continuous features, or intersections of the explicitly provided ones. Continuous features like age could be directly handled by tree-ensemble solvers. Besides, intersections of the input features could be captured by deeper tree structures. We designate the empirical performance and theoretical claims of these extensions for future research.

# Acknowledgment

ZSW was supported in part by the NSF award #2232692.

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

# A  Omitted Proofs in Section 4

We would first provide a formal definition of the discretization operation.

**Definition A.1** (Discretization Operation)**.** *A function $\tilde{f}_m$ is an $m$-discretized version of a predictor $f$ if*

- *there exists a monotonic non-decreasing function $d$ on $[0, 1]$ such that $d(f(x)) = \tilde{f}_m(x)$;*

- *$d$ is right-continuous on $[0, 1]$, that is*

$$\lim_{x \to c^+} d(x) = d(c), c \in [0, 1);$$

- *$|\mathcal{R}(\tilde{f}_m)| = m$, i.e., the range of $\tilde{f}_m$ is of size $m$.*

The additional requirement on right-continuity has minimal effect on the algorithm, but is only helpful for the proof of the following lemma.

**Lemma 4.2.** *Consider the post-processing step $p_{\mathcal{G}}$ in Equation (5). For all $f \in [0, 1]$ and $m \in \mathbb{N}$, and for any $m$-discretized version $\tilde{f}_m$ of $f$, we have*

$$\ell(p_{\mathcal{G}}(f), \mathcal{D}) \leq \ell(p_{\mathcal{G}}(\tilde{f}_m), \mathcal{D})$$

*Proof of Lemma 4.2.* Since

$$\ell(p_{\mathcal{G}}(f), \mathcal{D}) = \min_{T \subseteq \mathcal{T}(f, \mathcal{G})} \ell\left(f + \sum_{t \in T} t, \mathcal{D}\right),$$

we show that $\forall \epsilon > 0$, there exists $T \subseteq \mathcal{T}(f, \mathcal{G})$ such that

$$\ell\left(f + \sum_{t \in T} t, \mathcal{D}\right) \leq \ell\left(p_{\mathcal{G}}\left(\tilde{f}_m\right), \mathcal{D}\right) + \epsilon.$$

$$\ell\left(f + \sum_{t \in T} t, \mathcal{D}\right) - \ell\left(p_{\mathcal{G}}\left(\tilde{f}_m\right), \mathcal{D}\right)$$

$$= \mathbb{E}_{\mathcal{D}}\left[\left(f + \sum_{t \in T} t - y\right)^2\right] - \mathbb{E}_{\mathcal{D}}\left[\left(p_{\mathcal{G}}\left(\tilde{f}_m\right) - y\right)^2\right]$$

$$= \mathbb{E}_{\mathcal{D}}\left[\left(f + \sum_{t \in T} t - p_{\mathcal{G}}(\tilde{f}_m)\right)\left(f + \sum_{t \in T} t + p_{\mathcal{G}}(\tilde{f}_m) - 2y\right)\right]$$

$$\leq \sqrt{\mathbb{E}_{\mathcal{D}}\left[\left|f + \sum_{t \in T} t - p_{\mathcal{G}}(\tilde{f}_m)\right|^2\right]} \sqrt{\mathbb{E}_{\mathcal{D}}\left[\left(\left|f + \sum_{t \in T} t\right| + \left|p_{\mathcal{G}}(\tilde{f}_m)\right| + 2|y|\right)^2\right]}$$

$$\leq 4\sqrt{\mathbb{E}_{\mathcal{D}}\left[\left|f + \sum_{t \in T} t - p_{\mathcal{G}}(\tilde{f}_m)\right|^2\right]}.$$

Thus, it suffices to show that

$$\forall \epsilon > 0, \exists T \subseteq \mathcal{T}(f, \mathcal{G}), \left|f + \sum_{t \in T} t - p_{\mathcal{G}}(\tilde{f}_m)\right| \leq \epsilon/4 \tag{10}$$

Since $p_{\mathcal{G}}(\tilde{f}_m) = \tilde{f}_m + \sum_{t \in T'} t$ where $T' \in \mathcal{T}(\tilde{f}_m, \mathcal{G})$, we write

$$\left|f + \sum_{t \in T} t - p_{\mathcal{G}}(\tilde{f}_m)\right| \leq \left|f + \sum_{t \in T_1} t - \tilde{f}_m\right| + \left|\sum_{t \in T_2} t + \sum_{t \in T'} t\right|$$

and try to construct $T_1, T_2 \subseteq \mathcal{T}(f, \mathcal{G})$ to bound the two terms.

We first show the second term can be zero by showing that $\mathcal{T}(\tilde{f}_m, \mathcal{G}) \subseteq \mathcal{T}(f, \mathcal{G})$. By Equations (3) and (4), it suffices to show that

$$S_1\left(\tilde{f}_m(x)\right) \subseteq S_1(f), \tag{11}$$

that is,

$$\forall v \in [0, 1], \exists u \in [0, 1] \text{ s.t. } \{x \in \mathcal{X} : \tilde{f}_m(x) \geq v\} = \{x \in \mathcal{X} : f(x) \geq u\}$$

assuming $\{x \in \mathcal{X} : \tilde{f}_m(x) \geq v\} \neq \emptyset$.

Let

$$u = \min\{f(x) : \tilde{f}_m(x) \geq v\} \in [0, 1].^2$$

It follows directly that $\forall x$ s.t. $\tilde{f}_m(x) \geq v$, $f(x) \geq u$.

On the other hand, since $\tilde{f}_m = d(f)$ for some monotonically non-decreasing $d$, we have $\forall x$ s.t. $f(x) \geq u$, $\tilde{f}_m(x) = d(f(x)) \geq d(u)$. Since $u \in \{f(x) : \tilde{f}_m(x) \geq v\} = \{y : d(y) \geq v\}$, we have $d(u) \geq v$. Then $\{x \in \mathcal{X} : \tilde{f}_m(x) \geq v\} = \{x \in \mathcal{X} : f(x) \geq u\}$ follows directly. This indicates we can construct $T_2 = T'$.

It remains to show that

$$\forall \epsilon > 0, \exists T_1 \subseteq \mathcal{T}(f, \mathcal{G}) \text{ such that } \left| f + \sum_{t \in T_1} t - \tilde{f}_m \right| \leq \epsilon/4. \tag{12}$$

Let $u_i = \min\{f(x) : \tilde{f}_m(x) \geq v_i\}$ where $v_0 < \cdots < v_{m-1}$ is the range of $\tilde{f}_m$. We partition $[0, 1]$ into $\frac{2}{\epsilon}$ intervals

$$[0, \epsilon/2), [\epsilon/2, 2\epsilon/2), \cdots, [1 - \epsilon/2, 1].$$

For each $u_i$, we denote $n(u_i)$ as the largest integer such that $n(u_i) \cdot \epsilon/2 \leq u_i$. We further split the interval containing $u_i$ into two intervals $[n(u_i) \cdot \epsilon/2, u_i)$ and $[u_i, (n(u_i)+1) \cdot \epsilon/2)$. In this way, we partition $[0, 1]$ into $\frac{2}{\epsilon} + m - 1$ intervals such that each interval has size $\frac{\epsilon}{2}$, and that $u_1, \cdots, u_{m-1}$ can only be at the boundaries of some intervals. We write the intervals as $[l_1, r_1), \cdots, [l_{2/\epsilon+m-1}, r_{2/\epsilon+m-1}]$ where $l_1 = 0, r_{2/\epsilon+m-1} = 1$.

We then construct the following trees

$$c_i = d(l_i) - \frac{l_i + r_i}{2}, \quad t_i(x) = \mathbb{I}\{f(x) \geq l_i\} \cdot \left( c_i - \sum_{j=1}^{i-1} c_j \right), \quad T_1 = \{t_i\}_{i=1}^{2/\epsilon+m-1}$$

Then $\forall x$, assume $f(x) \in [l_k, r_k)$ for some $k$. Then

$$\left| f(x) + \sum_{t \in T} t(x) - \tilde{f}_m(x) \right|$$

$$= \left| f(x) + d(l_k) - \frac{l_k + r_k}{2} - d(l_k) \right|$$

$$= \left| f(x) - \frac{l_k + r_k}{2} \right| \leq \frac{r_k - l_k}{2} \leq \frac{\epsilon}{4}.$$

Therefore, we show that $\forall \epsilon > 0, \exists T_1 \subseteq \mathcal{T}(f, \mathcal{G})$ such that $\left| f + \sum_{t \in T_1} t - \tilde{f}_m \right| \leq \epsilon/4$. $\qquad \square$

---

[2]The minimum exists because the infimum belongs to the set: $\inf\{f(x) : \tilde{f}_m(x) \geq v\} \in \{f(x) : \tilde{f}_m(x) \geq v\}$. Recall that there exists a monotonic non-decreasing $d$ that's right-continuous on $[0, 1]$, s.t. $\tilde{f}_m(x) = d(f(x))$. It suffices to show that $y_0 \triangleq \inf\{y : d(y) \geq v\} \in \{y : d(y) \geq v\} \triangleq S$. By definition of infimum, $\forall n \in \mathbb{N}^*, \exists y_n \in S$ s.t. $y_0 \leq y_n < y_0 + \frac{1}{n}$. Then $\lim_{n\to\infty} y_n = y_0$. Since $d$ is right-continuous, $\lim_{n\to\infty} d(y_n) = d(y_0)$. Since $d(y_n) \geq v$, we have $d(y_0) \geq v$. Thus, $y_0 \in S$.

**Lemma 4.3.** *Given a predictor $f$ whose range is of size $m$, training an ensemble of decision trees on $f$ as in the post-processing step in Equation (5) encompasses adding a linear function of the groups on each output level set of $f$. That is,*

$$\left\{ f + \sum_{j=1}^{m} \mathbb{I}\{f(x) = v_j\} \left( c^j + \sum_i c_i^j g_i \right) \,\middle|\, c_i^j \in \mathbb{R} \right\} \subseteq \left\{ f + \sum_{t \in T} t \,\middle|\, T \subseteq \mathcal{T}(f, \mathcal{G}) \right\} \quad (7)$$

*where $v_1, \cdots v_m \in \mathcal{R}(f), v_1 < v_2 < \cdots < v_m$.*

*Proof of Lemma 4.3.* $\forall h \in \left\{ f + \sum_{j \in 1}^{m} \mathbb{I}\{f(x) = v_j\} \left( c^j + \sum_i c_i^j g_i \right) \,\middle|\, c_i^j \in \mathbb{R} \right\}$, let $h_j = c^j + \sum_i c_i^j g_i$. Then we can write $h$ as

$$h = f + \sum_{j \in 1}^{m} \mathbb{I}\{f(x) = v_j\} h_j$$

$$= f + \sum_{j \in 1}^{m} \mathbb{I}\{f(x) \geq v_j\} \left( h_j - \sum_{i=1}^{j-1} h_i \right)$$

$$\triangleq f + \sum_{j \in 1}^{m} \mathbb{I}\{f(x) \geq v_j\} h'_j,$$

where $h'_j$ is still a linear function of $g_i$. Let $h'_j = c'^j + \sum_i c'^j_i g_i$. By Equation (4),

$$\forall i, j, \ \mathbb{I}\{f \geq v_j\} c'^j_i g_i \in \mathcal{T}(f, \mathcal{G})$$

because we can set

$$s_1 = \{x \in \mathcal{X} : f(x) \geq v_{j-1}\}, s_2 = \{x \in \mathcal{X} : g_i(x) = 1\}, c_1 = c'^j_i, c_2 = c_3 = c_4 = 0.$$

And also,

$$\forall j, \ \mathbb{I}\{f \geq v_{j-1}\} c'^j \in \mathcal{T}(f, \mathcal{G})$$

because we can set

$$s_1 = \{x \in \mathcal{X} : f(x) \geq v_{j-1}\}, s_2 = \{x \in \mathcal{X} : g_1(x) = 1\}, c_1 = c_2 = c'^j, c_3 = c_4 = 0.$$

Thus, $\forall h \in \left\{ f + \sum_{j \in 1}^{m} \mathbb{I}\{f(x) = v_j\} \left( c^j + \sum_i c_i^j g_i \right) \,\middle|\, c_i^j \in \mathbb{R} \right\}$, we have

$$h \in \left\{ f + \sum_{t \in T} t \,\middle|\, T \subseteq \mathcal{T}(f, \mathcal{G}) \right\}.$$

$\square$

**Lemma 4.4.** *If a predictor $f$ with $\mathcal{R}(f) = \{v_1, \cdots, v_m\}$ has a multicalibration error w.r.t. $\mathcal{G}$ larger than $\alpha$, then there exists linear functions $h_j = \sum_i c_i^j g_i$, such that*

$$\ell(f, \mathcal{D}) > \ell \left( f + \sum_{j=1}^{m} \mathbb{I}\{f = v_j\} h_j, \mathcal{D} \right) + \alpha^2. \quad (8)$$

*Proof of Lemma 4.4.* Assume $f$ has the largest calibration error on the $k$-th group, namely

$$k = \arg\max_{i \in [|G|]} \sum_{v \in R(f)} \Pr_{(x,y) \sim \mathcal{D}} [f(x) = v, g_i(x) = 1] \cdot \left| \mathbb{E}_{(x,y) \sim \mathcal{D}} [f(x) - y \mid f(x) = v, g_i(x) = 1] \right|$$

Then we let

$$c_i^j = \begin{cases} -\mathbb{E}_{(x,y) \sim \mathcal{D}} [f(x) - y \mid f(x) = v_j, g_i(x) = 1], & \text{if } i = k; \\ 0, & \text{if } i \neq k. \end{cases} \quad (13)$$

Let $c_k^j = \alpha_j$. Then $h_j = \alpha_j g_k$. Then,

$$\ell(f, \mathcal{D}) - \ell\left(f + \sum_{j \in [|R(f)|]} \mathbb{I}\{f = v_j\}h_j, \mathcal{D}\right)$$

$$= \sum_{j=1}^{m} \Pr_{(x,y)\sim\mathcal{D}}[f(x) = v_j]\mathbb{E}_{\mathcal{D}}\left[(f(x) - y)^2 \mid f(x) = v_j\right]$$

$$- \sum_{j=1}^{m} \Pr_{(x,y)\sim\mathcal{D}}[f(x) = v]\mathbb{E}_{\mathcal{D}}\left[(f(x) + h_j(x) - y)^2 \mid f(x) = v_j\right]$$

$$= - \sum_{j=1}^{m} \Pr_{(x,y)\sim\mathcal{D}}[f(x) = v_j]\mathbb{E}_{\mathcal{D}}\left[2h_j(x)(f(x) - y) + h_j^2(x) \mid f(x) = v_j\right]$$

$$= - \sum_{j=1}^{m} \Pr_{(x,y)\sim\mathcal{D}}[f(x) = v_j]\mathbb{E}_{\mathcal{D}}\left[2\alpha_j g_k(f(x) - y) + \alpha_j^2 g_k \mid f(x) = v_j\right]$$

$$= - \Pr_{(x,y)\sim\mathcal{D}}[g_k(x) = 1] \sum_{j=1}^{m} \Pr_{(x,y)\sim\mathcal{D}}[f(x) = v_j]\mathbb{E}_{\mathcal{D}}\left[2\alpha_j(f(x) - y) + \alpha_j^2 \mid f(x) = v_j, g_k(x) = 1\right]$$

$$= \Pr_{(x,y)\sim\mathcal{D}}[g_k(x) = 1] \sum_{j=1}^{m} \Pr_{(x,y)\sim\mathcal{D}}[f(x) = v_j]\alpha_j^2$$

The final equation is due to $\alpha_j = -\mathbb{E}_{(x,y)\sim\mathcal{D}}\left[f(x) - y \mid f(x) = v_j, g_i(x) = 1\right]$. By the definition of multicalibration error, Equation (13) indicates that

$$\sum_{j=1}^{m} \Pr_{(x,y)\sim\mathcal{D}}[f(x) = v_j]|\alpha_j| > \frac{\alpha}{\Pr_{(x,y)\sim\mathcal{D}}[g_k(x) = 1]}.$$

By Cauchy-Schwarz inequality, we have

$$\ell(f, \mathcal{D}) - \ell\left(f + \sum_{j \in [|R(f)|]} \mathbb{I}\{f = v_j\}h_j, \mathcal{D}\right)$$

$$= \Pr_{(x,y)\sim\mathcal{D}}[g_k(x) = 1] \sum_{j=1}^{m} \Pr_{(x,y)\sim\mathcal{D}}[f(x) = v_j]\alpha_j^2$$

$$\geq \Pr_{(x,y)\sim\mathcal{D}}[g_k(x) = 1] \frac{\left(\sum_{j=1}^{m} \Pr_{(x,y)\sim\mathcal{D}}[f(x) = v_j]\alpha_j\right)^2}{\sum_{j=1}^{m} \Pr_{(x,y)\sim\mathcal{D}}[f(x) = v_j]}$$

$$> \frac{\alpha^2}{\Pr_{(x,y)\sim\mathcal{D}}[g_k(x) = 1]} \geq \alpha^2.$$

$\square$

## B   Discussion on the applicability of the algorithm

Our algorithm consistently achieved low multicalibration error across all the datasets we experimented with, demonstrating its effectiveness in realistic datasets. However, to the best of our knowledge, we could not find a guarantee that our algorithm will always return a model with low multicalibration error. The main difficulty is to obtain a guarantee that Assumption 4.5 holds.

On the other hand, it's possible to construct an example where our algorithm fails to achieve low multicalibration error. Unlike the above experiments where we post process upon a predictor, the construction uses an unrealistic setting where the base predictor outputs a constant value. Assume

the goal is to obtain multicalibration w.r.t. three binary groups, namely $\boldsymbol{g}(x) \in \{0,1\}^3$, with $g_i(x) = 0$ w.p. 0.5, and $g_i$'s being independent. The ground truth $y(x)$ is given by

$$y(x) = (1-\gamma)\left(\frac{1}{2}g_1(x) + \frac{1}{4}g_2(x) + \frac{1}{8}g_3(x)\right)$$
$$+ \gamma(g_1(x) \oplus g_2(x) \oplus g_3(x)),$$

where $\gamma \in (0,1)$ is a constant and $\oplus$ denotes the XOR operation. Note that the base predictor is a constant, so a loss minimizer given by our algorithm is

$$f(x) = (1-\gamma)\left(\frac{1}{2}g_1(x) + \frac{1}{4}g_2(x) + \frac{1}{8}g_3(x)\right) + \frac{\gamma}{2},$$

as an ensemble of decision trees of depth 2 will never be able to capture the XOR operation. The first term makes sure that $f(x_i) \neq f(x_j)$ whenever $\boldsymbol{g}(x_i) \neq \boldsymbol{g}(x_j)$, leaving a multicalibration error

$$\sum_{v \in R(f)} \Pr_{(x,y)\sim\mathcal{D}}[f(x) = v, g_i(x) = 1] \cdot$$
$$\left|\mathbb{E}_{(x,y)\sim\mathcal{D}}\left[f(x) - y \mid f(x) = v, g_i(x) = 1\right]\right|$$
$$= \sum_{v \in R(f)} \Pr_{(x,y)\sim\mathcal{D}}[f(x) = v, g_i(x) = 1] \cdot \frac{\gamma}{2} = \frac{\gamma}{4}.$$

In this example, $\ell(f, \mathcal{D}) = \gamma^2/4$, and since $f(x)$ outputs a different value for different value of $\boldsymbol{g}(x)$, running our algorithm a second time gives a zero-error $p_{\mathcal{G}}(f) = y(x)$, making $\epsilon_{loss} = \gamma^2/4$. This large improvement is a violation of Assumption 4.5, which explains why the algorithm is not able to achieve low multicalibration error in this case. This is mainly due to the lack of information in the base predictor, rendering the ensemble less useful. Again, we emphasize that this is construction is often unrealistic as we usually have a non-trivial base predictor that gives meaningful prediction before we seek to calibrate it for fairness or robustness across distributions.

## C   Implementation of the trees in established tree-ensemble solvers

In Equation (4), we defined the set of trees $\mathcal{T}$ of our algorithm, where we require that each tree splits on a single threshold and a single group. However, this might not be what solvers like Ke et al. (2017) and Chen & Guestrin (2016) implement. In fact, with $f_0(x)$ and $g_i$ given as the input, depth-2 trees in those solvers are given by

$$\mathcal{T}'(f_0, \mathcal{G}) = \Big\{ c_1 \cdot \mathbb{I}\{x \in s_1 \wedge x \in s_2\} + c_2 \cdot \mathbb{I}\{x \in s_1 \wedge x \notin s_2\} + c_3 \cdot \mathbb{I}\{x \notin s_1 \wedge x \in s_3\}$$
$$+ c_4 \cdot \mathbb{I}\{x \notin s_1 \wedge x \notin s_3\} : c_1, c_2, c_3, c_4 \in \mathbb{R}, s_1, s_2, s_3 \in \mathcal{S}_1(f_0) \cup \mathcal{S}_2(\mathcal{G}) \Big\} \tag{14}$$

where $\mathcal{S}_1$ and $\mathcal{S}_2$ are the same as Equation (3). We want to show that our arguments in Section 4 still hold for such implementation.

From Equation (14) we see that $\mathcal{T} \subseteq \mathcal{T}'$. If $\mathcal{T}$ is substituted by $\mathcal{T}'$, Lemma 4.2 would still be valid, as guaranteed by Equation (11) and Equation (12), which still hold for the expanded set $\mathcal{T}'$. Lemma 4.3 would still be valid as well, as

$$\left\{ f + \sum_{t \in T} t \ \middle|\ T \subseteq \mathcal{T}(f, \mathcal{G}) \right\} \subseteq \left\{ f + \sum_{t \in T} t \ \middle|\ T \subseteq \mathcal{T}'(f, \mathcal{G}) \right\}.$$

Lemma 4.4 does not involve the definition of $\mathcal{T}$. Therefore, the analysis in Section 4 applies to these established solvers.

## D   Description of Datasets

In this section we describe the datasets we use for evaluation the proposed method.

Table 4: The number of groups defined for the ACS dataset.

| Attribute | Number of groups defined on the attribute | | |
|---|---|---|---|
| | Income Regression | Income Classification | Travel Time Regression |
| Race | 5 | 5 | 5 |
| Gender | 2 | 2 | 2 |
| Age | 5 | 5 | 5 |
| Education | 6 | 6 | 6 |
| Occupation | 30 | 30 | 33 |
| Total | 48 | 48 | 51 |

## D.1 ACS Dataset

The American Community Survey (ACS) dataset is a public dataset that contains demographic information about the US population, and can be accessed through the `folktables` package (Ding et al., 2021). In this work we consider three tasks using data retrieved from this dataset, namely Income Regression, Income Classification, and Travel Time Regression. We used the subset of the 2018 census data in California.

**Data preprocess** `Folktables` already defines the features associated with each task and provides a preprocessing filter that extracts a set of reasonably clean records. We used the provided filter to preprocess the data. Additionally, for income regression task, since the income has a long tail, scaling the income value to $[0, 1]$ would lead to a large mass of data points around 0. As implemented in Globus-Harris et al. (2023), we filter out the data points with income higher than 100,000 and scale the income to $[0, 1]$. Travel time regression does not have a similar long tail, so we directly scale the travel time to $[0, 1]$.

**Dataset partition and the uncalibrated base predictors** Given the features provided in `Folktables` that does not contain `nan` features, we fit a linear regressor for regression tasks and a linear SVM model for classification tasks using randomly sampled 50,000 data entries. Apart from that, we set aside 12,000 data entries for the test set and 18,000 entries for the calibration and validation set.

**Groups** In order to create a rich set of groups, we construct the binary groups using the five attributes in the dataset: race, gender, age, education and occupation. We transformed the categorized features into one-hot encoding. Specifically, we used age groups instead of the numerical age, and we categorized the occupation by the first two digits of the occupation codes. For each category, as long as it contains more than 1% of the data, we accept it as a group. In this way, we defined around 50 groups for each of the three tasks. (See Table 4)

## D.2 CivilComments Dataset

This dataset came from the `WILDS` dataset(Koh et al., 2021; Borkan et al., 2019) which collected 447,998 comments labeled by crowd workers. The task is to predict whether a comment is considered toxic or not.

**Dataset partition and the uncalibrated base predictor** Koh et al. (2021) partitioned the dataset into train set (60%), validation set (10%) and test set (30%), and provided a baseline predictor. We set aside one-sixth of the training set and combine it withe validation set for calibration and validation. As for the base predictor, we also followed the training of the baseline algorithm in the original paper and trained a DistilBERT (Sanh et al., 2020) model using the same specified hyperparameters and settings.

**Groups** Koh et al. (2021) provides eight predefined groups based on the identity terms mentioned in the comments, namely male, female, LGBTQ, Christian, Muslim, other religions, black, and white. We use these groups for the multicalibration evaluation.

### D.3 UTKFace Dataset

The UTKFace dataset (Zhang et al., 2017) consists of 23,707 face images with annotations of age, gender and race, ranging from 1 to 120 years old. We use the dataset for the age regression task.

**Dataset partition and the uncalibrated base predictor**   We used 40% of the data for training, 40% for calibration and validation, and 20% for testing. The base predictor was trained on the training set with a ResNet-18 model for 30 epochs with early stopping enabled.

**Groups**   We defined 16 groups (2 for gender, 5 for race, and 9 for age) on this dataset.

### D.4 ISIC Dataset

The International Skin Imaging Collaboration (ISIC) dataset is a public dataset that contains images of skin lesions together with patient metadata and diagnosis label. We use the data provided in the ISIC 2019 challenge(Tschandl et al., 2018; Codella et al., 2017; Combalia et al., 2019). The original task is to classify the skin lesion images into one of the seven categories, and we consider the derived binary classification task for predicting the category NV, which accounts for 50.8% of the dataset.

**Dataset partition and the uncalibrated base predictor**   Since the test labels are not made public, we only use the 25,331 training images. The images in the dataset come from three sources: 49.0% from BCN (Combalia et al., 2019), 39.5% from HAM (Tschandl et al., 2018), and 11.5% from Codella et al. (2017) or other sources. We use the 12413 BCN images as the training set for the base predictor, three-fourths of the HAM images (7511 images) as the calibration and validation set, and the remaining 5407 images as the test set. The base predictor was trained on the training set with a ResNet-18 multi-classification model for 20 epochs with early stopping enabled. The predicted probability of the category NV was used as the base predictor output.

**Groups**   The ISIC challenge provides metadata, which includes approximate age and gender of the patients as well as the anatomy site. Just like in the ACS dataset, we consider a category as a group if it has at least 1% of the data. In total 14 binary groups (2 for gender, 5 for anatomy site, and 7 for age) were defined.

## E   Details of calibration algorithms

In our experiments, we fix the uncalibrated base predictor and the test set and introduce randomness by partitioning the remaining part into an equal-sized calibration and validation set 10 times. After finding the best hyperparameters on the 10 folds, we calibrate the base predictor using this specific choice of hyperparameters and report the performance on the test set. For LSBoost and MCBoost, since the number of level sets $m$ needs to be determined at calibration time, we sweep over the hyperparameters on the ten folds for all 6 level sets we evaluated, namely $\{10, 20, 30, 50, 75, 100\}$, and choose the best hyperparameters for each level set.

We now introduce the algorithms' implementation detail and the hyperparameters involved:

**Multiaccurate Baseline**   It's been shown in Roth (2022) that a linear model with $\ell_1$ regularization guarantees multiaccuracy. We use the binary groups as the feature and sweep over the regularization strength $\lambda$ in $\{0, 10^{-6}, 10^{-5}, \cdots, 10^{-2}\}$.

**Ours**   Our algorithm works pretty much out-of-the-box by passing the uncalibrated prediction, the groups and the ground truth to the `LightGBM`(Ke et al., 2017) solver. To avoid the complexity of determining the optimal number of trees, we use early stopping with patience of 50 iterations and set a high maximum limit of 5000 trees. 30% of the calibration set is set aside during calibration to monitor if the loss continues to decrease, and this is different from the validation set, which is solely used to tune the hyperparameters. In other words, given a set of hyperparameters, the calibration set is the only data that is used, though internally, only 70% of the calibration set is used for learning the parameters. The depth of each tree is two, as indicated in the main body. We vary the learning rate across five exponentially spaced points from 0.01 to 1 and adjust the feature subsampling ratio linearly across 10 points from 0.1 to 1.

**LSBoost**  LSBoost is introduced in Globus-Harris et al. (2023), and we used the implementation provided by the authors, with minor modifications in the multiprocessing implementation and rounding the output to the $1/m$ grids. Unlike the algorithm described in the original paper, the authors also implemented early stopping in their code base, and for better performance, this is also enabled in our experiment with 30% of the calibration set as described in the previous paragraph. Considering that our algorithm uses decision trees of depth 2 as the weak learners, we experimented with trees of depth both 1 and 2 during hyperparameter sweep. Other hyperparameters include the learning rate, which can be 0.1, 0.3, or 1, and the subsampling ratio for the weak learner, which is adjusted linearly across 10 points from 0.1 to 1. In total, $2 \times 3 \times 10 = 60$ hyperparameter combinations are evaluated on all ten folds and all six calibration level sets.

**MCBoost**  We implement the algorithm described in Roth (2022), which is a variant of the primitive multicalibration algorithm proposed by Hebert-Johnson et al. (2018). This algorithm doesn't involve any hyperparameters, but it's known that it can be prone to overfit, so we also set aside a proportion of the calibration set for early stopping. We consider this proportion as a hyperparameter, and swept over the values $\{0.1, 0.2, 0.3, 0.4, 0.5\}$. Like LSBoost, the hyperparameters are evaluated on all ten folds and all six calibration level sets.

The uncalibrated predictors for the image or text task were trained on an A100 GPU or an RTX 4090 GPU. Other predictors, as well as the calibrators, were trained on an AMD 64-core CPU. Running all the experiments takes approximately 6 hours.

# F  Statistical Learning Guarantees for Tree Ensemble Optimization

In Section 4, we established a theoretical framework that connects multicalibration error with loss, demonstrating that when decision tree ensembles achieve effective loss minimization, the multicalibration error can be bounded. However, in the finite sample regime, there would be an additional excess risk term due to the empirical (instead of distributional) loss minimization. This section extends our analysis, examining how a finite sample affects these theoretical guarantees.

Recall that we constructed the trees in the ensemble using the splitting criteria defined in Equation (3)

$$\mathcal{S}_1(f_0) = \left( \bigcup_{v \in \mathcal{R}(f_0) \cup \{0\}} \{x \in \mathcal{X} : f_0(x) \geq v\} \right),$$

$$\mathcal{S}_2(\mathcal{G}) = \left( \bigcup_{i \in [|\mathcal{G}|]} \{x \in \mathcal{X} : g_i(x) = 1\} \right),$$

Each tree in $\mathcal{T}(f_0, \mathcal{G})$ can be formally written as

$$c_1 \cdot \mathbb{I}\{x \in s_1 \wedge x \in s_2\} + c_2 \cdot \mathbb{I}\{x \in s_1 \wedge x \notin s_2\} + c_3 \cdot \mathbb{I}\{x \notin s_1 \wedge x \in s_2\} + c_4 \cdot \mathbb{I}\{x \notin s_1 \wedge x \notin s_2\}$$

where $s_1 \in \mathcal{S}_1(f_0)$, and $s_2 \in \mathcal{S}_2(\mathcal{G})$. Here we can require $c_1, c_2, c_3, c_4 \in [-1, 1]$, because the ground truth label $\mathcal{Y}$ is in $[0, 1]$.

As shown in equation 5, the post-processed predictor is obtained by solving

$$p_{\mathcal{G}}(f_0) = \underset{T \subseteq \mathcal{T}(f_0, \mathcal{G})}{\arg\min} \ \ell\left(f_0 + \sum_{t \in T} t, \mathcal{D}\right)$$

With only a finite number of samples, the excess risk measures the discrepancy between the loss of the empirical predictor $\hat{p}_{\mathcal{G}}(f_0)$ and the optimal one on the distribution $p_{\mathcal{G}}(f_0)$

$$\epsilon_{excess} = \ell\left(\hat{p}_{\mathcal{G}}(f_0), \mathcal{D}\right) - \ell\left(p_{\mathcal{G}}(f_0), \mathcal{D}\right)$$

With this term, the multicalibration error $\alpha$ in Theorem 4.6 would change to be

$$\alpha \leq \sqrt{\epsilon_{loss} + \epsilon_{round} + \epsilon_{excess}}$$

$\epsilon_{loss}$ only depends on the distribution, and is assumed to be small according to Assumption 4.5. We also do not focus on the case where the rounding error $\epsilon_{round}$ dominates neither. In this section, we try to derive a bound on the excess risk $\epsilon_{excess}$ by analyzing the convergence and the generalization of loss minimization on our ensemble. We then try to provide an asymptotic analysis of how many samples are required to achieve a target multicalibration error $\alpha$ by considering $\alpha = O(\sqrt{\epsilon_{excess}})$.

## F.1 Loss minimization in practice

Loss minimization is typically done through adding trees to the ensemble

$$p_{\mathcal{G}}(f_0) = \underset{T \subseteq \mathcal{T}(f_0, \mathcal{G})}{\arg \min} \ \ell \left( f_0 + \sum_{t \in T} t, \mathcal{D} \right)$$

In practice, we typically limit the number of trees added, namely the size of subset $T$, by setting a maximum limit or through early stopping. This keeps the running time of the algorithm reasonable, and can balance optimization error reduction against model complexity. This motivates the introduction of a parameter $N_T$ to represent the maximum number of trees in our ensemble. We define the function class $\mathcal{F}_{N_T}$ that we empirically optimize over as

$$\mathcal{F}_{N_T} = \left\{ f_0 + \sum_{i=1}^{n} t_i \ \middle| \ n \le N_T, t_i \in \mathcal{T}(f_0, \mathcal{G}) \right\}$$

## F.2 Rademacher Complexity of $\mathcal{F}_{N_T}$

The generalization error increases as $N_T$ increases and as the model becomes more complex. We use the Rademacher complexity to quantify the complexity so as to bound the generalization error using the number of samples, the number of trees, and the number of groups. We first recall the definition of the Rademacher complexity of a class of functions

**Definition F.1.** *For a hypothesis class $\mathcal{H}$ and a sample size $n$, the Rademacher complexity $R_n(\mathcal{H})$ is defined as*

$$R_n(\mathcal{H}) = \mathbb{E}_{\sigma, S}[\sup_{h \in \mathcal{H}} \frac{1}{n} \sum_{i=1}^{n} \sigma_i h(x_i)]$$

*where $S = \{x_1, x_2, ..., x_n\}$ is a sample of $n$ points drawn i.i.d. from distribution $\mathcal{D}$, and $\sigma = \{\sigma_1, \sigma_2, ..., \sigma_n\}$ are independent Rademacher random variables (taking values $+1$ or $-1$ with equal probability).*

We now state several key lemmas that will be used in our analysis. These lemmas are standard results in statistical learning theory and can be found in Mohri et al. (2018).

**Lemma F.2** (Massart's Lemma). *Let $\mathcal{A} \subseteq \mathbb{R}^m$ be a finite set, with $r = \max_{\mathbf{x} \in \mathcal{A}} \|\mathbf{x}\|_2$, then the following holds*

$$\mathbb{E}_{\boldsymbol{\sigma}} \left[ \frac{1}{m} \sup_{\mathbf{x} \in \mathcal{A}} \sum_{i=1}^{m} \sigma_i x_i \right] \le \frac{r \sqrt{2 \log |\mathcal{A}|}}{m}$$

*where $\sigma_i s$ are independent uniform random variables taking values in $\{-1, +1\}$ and $x_1, \ldots, x_m$ are the components of vector $\mathbf{x}$.*

**Lemma F.3** (Talagrand's Contraction Lemma). *Let $\mathcal{H}$ be a class of real-valued functions. If $\phi : \mathbb{R} \to \mathbb{R}$ is a Lipschitz function with Lipschitz constant L, then*

$$\mathbb{E}_{\sigma} \left[ \sup_{h \in \mathcal{H}} \frac{1}{n} \sum_{i=1}^{n} \sigma_i \phi(h(x_i)) \right] \le L \cdot \mathbb{E}_{\sigma} \left[ \sup_{h \in \mathcal{H}} \frac{1}{n} \sum_{i=1}^{n} \sigma_i h(x_i) \right]$$

**Lemma F.4** (Sum Rule). *For any two hypothesis sets $\mathcal{H}$ and $\mathcal{H}'$ of functions mapping from $\mathcal{X}$ to $\mathbb{R}$*

$$R_m(\mathcal{H} + \mathcal{H}') = R_m(\mathcal{H}) + R_m(\mathcal{H}')$$

*where $\mathcal{H} + \mathcal{H}' = \{h + h' : h \in \mathcal{H}, h' \in \mathcal{H}'\}$ is the sum class.*

To analyze the Rademacher complexity of our tree ensemble function class $\mathcal{F}_{N_T}$, we begin by analyzing the components that make up our trees. We introduce the following auxiliary function classes

- $\mathcal{F}_1 = \{\mathbb{I}\{x \in s_1\}, \mathbb{I}\{x \notin s_1\} \mid s_1 \in \mathcal{S}_1(f_0)\}$: the set of threshold indicator functions $\mathbb{I}\{x \in s_1\} = \mathbb{I}\{f_0(x) \geq v\}$ for $v \in \mathcal{R}(f_0) \cup \{0\}$ and their complements $\mathbb{I}\{x \notin s_1\} = \mathbb{I}\{f_0(x) < v\}$
- $\mathcal{F}_2 = \{\mathbb{I}\{x \in s_2\}, \mathbb{I}\{x \notin s_2\} \mid s_2 \in \mathcal{S}_2(\mathcal{G})\}$: the set of group indicator functions $\mathbb{I}\{x \in s_2\} = \mathbb{I}\{g_i(x) = 1\}$ for $i \in [|\mathcal{G}|]$ and their complements $\mathbb{I}\{x \notin s_2\} = \mathbb{I}\{g_i(x) = 0\}$

We proceed through the following steps

**Lemma F.5.** *For the threshold indicator function class* $\mathcal{F}_1 = \{\mathbb{I}\{x \in s_1\}, \mathbb{I}\{x \notin s_1\} \mid s_1 \in \mathcal{S}_1(f_0)\}$, *the Rademacher complexity is*

$$R_n(\mathcal{F}_1) = O(n^{-1/2})$$

*Proof.* Let $S = \{x_1, \ldots, x_n\}$ be a sample, and let $\sigma_1, \ldots, \sigma_n$ be Rademacher random variables. Consider the values $f_0(x_1), \ldots, f_0(x_n)$ and rearrange them in non-decreasing order as $f_0(x_{(1)}) \leq \ldots \leq f_0(x_{(n)})$.

For any threshold $v \in \mathcal{R}(f_0) \cup \{0\}$, there exists a $k \in \{0, 1, \ldots, n\}$ such that $\mathbb{I}\{f_0(x_{(i)}) \geq v\} = 1$ for $i > k$ and $\mathbb{I}\{f_0(x_{(i)}) \geq v\} = 0$ for $i \leq k$. Thus,

$$\sum_{i=1}^{n} \sigma_i \mathbb{I}\{f_0(x_i) \geq v\} = \sum_{i=k+1}^{n} \sigma_{(i)}$$

where $\sigma_{(i)}$ is the Rademacher variable corresponding to $x_{(i)}$.

Similarly, for the complement indicator

$$\sum_{i=1}^{n} \sigma_i \mathbb{I}\{f_0(x_i) < v\} = \sum_{i=1}^{k} \sigma_{(i)}$$

Therefore

$$\sup_{f \in \mathcal{F}_1} \sum_{i=1}^{n} \sigma_i f(x_i) = \max\left\{ \max_{0 \leq k \leq n} \left|\sum_{i=1}^{k} \sigma_{(i)}\right|, \max_{0 \leq k \leq n} \left|\sum_{i=k+1}^{n} \sigma_{(i)}\right| \right\}$$

Since the Rademacher variables are symmetric, both terms have the same distribution, and we have

$$\mathbb{E}_\sigma\left[ \sup_{f \in \mathcal{F}_1} \frac{1}{n} \sum_{i=1}^{n} \sigma_i f(x_i) \right] = \frac{1}{n} \mathbb{E}_\sigma\left[ \max_{0 \leq k \leq n} \left|\sum_{i=1}^{k} \sigma_i\right| \right]$$

We now need to bound $\mathbb{E}_\sigma[\max_{0 \leq k \leq n} |\sum_{i=1}^{k} \sigma_i|]$. Let $S_k = \sum_{i=1}^{k} \sigma_i$. Using Doob's martingale inequality, for any $\lambda > 0$

$$P\left( \max_{0 \leq k \leq n} |S_k| \geq \lambda \right) \leq \frac{\mathbb{E}[S_n^2]}{\lambda^2} = \frac{n}{\lambda^2}$$

Using the formula for the expectation of a non-negative random variable

$$\mathbb{E}\left[ \max_{0 \leq k \leq n} |S_k| \right] = \int_0^\infty P\left( \max_{0 \leq k \leq n} |S_k| \geq t \right) dt$$

$$\leq \int_0^\infty \min\left(1, \frac{n}{t^2}\right) dt$$

$$= \int_0^{\sqrt{n}} 1 \, dt + \int_{\sqrt{n}}^\infty \frac{n}{t^2} dt$$

$$= \sqrt{n} + \left(-\frac{n}{t}\right)\Bigg|_{\sqrt{n}}^\infty$$

$$= \sqrt{n} + \sqrt{n} = 2\sqrt{n}$$

Therefore

$$\mathbb{E}_\sigma \left[ \max_{0 \le k \le n} \left| \sum_{i=1}^{k} \sigma_i \right| \right] \le 2\sqrt{n}$$

And

$$R_n(\mathcal{F}_1) = \frac{1}{n} \mathbb{E}_\sigma \left[ \max_{0 \le k \le n} \left| \sum_{i=1}^{k} \sigma_i \right| \right] \le \frac{2\sqrt{n}}{n} = \frac{2}{\sqrt{n}} = O(n^{-1/2})$$

$\square$

**Lemma F.6.** *For the group indicator function class $\mathcal{F}_2 = \{\mathbb{I}\{x \in s_2\}, \mathbb{I}\{x \notin s_2\} \mid s_2 \in \mathcal{S}_2(\mathcal{G})\}$, the Rademacher complexity is*

$$R_n(\mathcal{F}_2) = O\left( \sqrt{\frac{\log(|\mathcal{G}|)}{n}} \right)$$

*Proof.* We prove by applying Lemma F.2 (Massart's Lemma). We first identify the set $\mathcal{A}$ and compute the parameter $r$ in the lemma.

Let $S = \{x_1, \dots, x_n\}$ be a sample. For each function $f \in \mathcal{F}_2$, we can represent it as a vector $\mathbf{x}_f = (f(x_1), \dots, f(x_n)) \in \{0, 1\}^n \subset \mathbb{R}^n$. Let $\mathcal{A} = \{\mathbf{x}_f : f \in \mathcal{F}_2\}$ be the set of all such vectors.

Note that $|\mathcal{A}| = |\mathcal{F}_2| \le 2|\mathcal{G}|$, since $\mathcal{F}_2$ consists of at most $|\mathcal{G}|$ group indicator functions and their complements.

For $\mathbf{x}_f \in \mathcal{A}$, since $f(x_i) \in \{0, 1\}$ for all $i$, we have

$$\|\mathbf{x}_f\|_2 = \sqrt{\sum_{i=1}^{n} f(x_i)^2} = \sqrt{\sum_{i=1}^{n} f(x_i)} \le \sqrt{n}$$

Therefore, $r = \max_{\mathbf{x} \in \mathcal{A}} \|\mathbf{x}\|_2 \le \sqrt{n}$.

By Massart's Lemma

$$\begin{aligned}
\mathbb{E}_{\boldsymbol{\sigma}} \left[ \frac{1}{n} \sup_{\mathbf{x} \in \mathcal{A}} \sum_{i=1}^{n} \sigma_i x_i \right] &\le \frac{r \sqrt{2 \log |\mathcal{A}|}}{n} \\
&\le \frac{\sqrt{n} \cdot \sqrt{2 \log(2|\mathcal{G}|)}}{n} \\
&= \frac{\sqrt{2 \log(2|\mathcal{G}|)}}{\sqrt{n}} \\
&= O\left( \sqrt{\frac{\log(|\mathcal{G}|)}{n}} \right)
\end{aligned}$$

By definition, this expectation is precisely the Rademacher complexity $R_n(\mathcal{F}_2)$, thus completing the proof. $\square$

**Lemma F.7.** *For the product class consisting of all possible products of indicators from $\mathcal{F}_1$ and $\mathcal{F}_2$, denoted as $\mathcal{F}_1 \otimes \mathcal{F}_2 = \{f_1 \cdot f_2 \mid f_1 \in \mathcal{F}_1, f_2 \in \mathcal{F}_2\}$, the Rademacher complexity is*

$$R_n(\mathcal{F}_1 \otimes \mathcal{F}_2) = O\left( \sqrt{\frac{\log(|\mathcal{G}|)}{n}} \right)$$

*Proof.* For any $f_1 \in \mathcal{F}_1$ and $f_2 \in \mathcal{F}_2$, we have the identity

$$f_1 \cdot f_2 = \frac{1}{2}[(f_1 + f_2)^2 - f_1^2 - f_2^2].$$

Therefore, the product class $\mathcal{F}_1 \otimes \mathcal{F}_2$ is a subset of

$$\frac{1}{2}[(\mathcal{F}_1 + \mathcal{F}_2)^2 + (-1) \cdot \mathcal{F}_1^2 + (-1) \cdot \mathcal{F}_2^2],$$

where $\mathcal{F}^2 = \{f^2 : f \in \mathcal{F}\}$ and $(\mathcal{F}_1 + \mathcal{F}_2)^2 = \{(f_1 + f_2)^2 : f_1 \in \mathcal{F}_1, f_2 \in \mathcal{F}_2\}$.

Using the properties of Rademacher complexity, we can obtain

$$R_n(\mathcal{F}_1 \otimes \mathcal{F}_2) \le \frac{1}{2}[R_n((\mathcal{F}_1 + \mathcal{F}_2)^2) + R_n(\mathcal{F}_1^2) + R_n(\mathcal{F}_2^2)].$$

By Talagrand's Contraction Lemma, for the squaring operation (which is Lipschitz with constant 2 when the norms of the function output are bounded by 1)

$$R_n((\mathcal{F}_1 + \mathcal{F}_2)^2) \le 2 \cdot R_n(\mathcal{F}_1 + \mathcal{F}_2) = 2 \cdot [R_n(\mathcal{F}_1) + R_n(\mathcal{F}_2)]$$

Similarly,

$$R_n(\mathcal{F}_1^2) \le 2 \cdot R_n(\mathcal{F}_1), \quad R_n(\mathcal{F}_2^2) \le 2 \cdot R_n(\mathcal{F}_2)$$

Substituting these bounds give

$$R_n(\mathcal{F}_1 \otimes \mathcal{F}_2) \le \frac{1}{2}[2(R_n(\mathcal{F}_1) + R_n(\mathcal{F}_2)) + 2R_n(\mathcal{F}_1) + 2R_n(\mathcal{F}_2)] = 2R_n(\mathcal{F}_1) + 2R_n(\mathcal{F}_2)$$

Given that $R_n(\mathcal{F}_1) = O(n^{-1/2})$ and $R_n(\mathcal{F}_2) = O(\sqrt{\log(|\mathcal{G}|)/n})$, when $|\mathcal{G}| > 1$, the dominant term is $R_n(\mathcal{F}_2)$. Therefore,

$$R_n(\mathcal{F}_1 \otimes \mathcal{F}_2) = O\left(\sqrt{\frac{\log(|\mathcal{G}|)}{n}}\right).$$

$\square$

**Proposition F.8.** *The Rademacher complexity of the tree ensemble function class $\mathcal{F}_{N_T}$ is bounded by*

$$R_n(\mathcal{F}_{N_T}) = O\left(N_T \sqrt{\frac{\log(|\mathcal{G}|)}{n}}\right)$$

*Proof.* First, we consider the Rademacher complexity for the class of trees with 4 leaves, denoted as $\mathcal{T}(f_0, \mathcal{G})$. Each tree in $\mathcal{T}(f_0, \mathcal{G})$ can be expressed as a linear combination of 4 indicator products, corresponding to the 4 leaves

$$c_1 \cdot \mathbb{I}\{x \in s_1 \wedge x \in s_2\} + c_2 \cdot \mathbb{I}\{x \in s_1 \wedge x \notin s_2\} + c_3 \cdot \mathbb{I}\{x \notin s_1 \wedge x \in s_2\} + c_4 \cdot \mathbb{I}\{x \notin s_1 \wedge x \notin s_2\}$$

Let $\mathcal{H}_1 = \{c_1 \cdot \mathbb{I}\{x \in s_1 \wedge x \in s_2\} : s_1 \in \mathcal{S}_1(f_0), s_2 \in \mathcal{S}_2(\mathcal{G}), c_1 \in [-1, 1]\}$, and similarly define $\mathcal{H}_2$, $\mathcal{H}_3$, and $\mathcal{H}_4$ for the other terms. The indicator products like $\mathbb{I}\{x \in s_1 \wedge x \in s_2\}$ are elements of $\mathcal{F}_1 \otimes \mathcal{F}_2$. By Talagrand's Contraction Lemma (Lemma F.3), since the coefficients $c_i \in [-1, 1]$, they are bounded by $C_{max} = 1$. The function $\phi(h) = c \cdot h$ for a fixed $c \in [-1, 1]$ is Lipschitz with constant $|c| \le 1$. Therefore, for each term $j \in \{1, 2, 3, 4\}$

$$R_n(\mathcal{H}_j) \le 1 \cdot R_n(\mathcal{F}_1 \otimes \mathcal{F}_2) = O\left(\sqrt{\frac{\log(|\mathcal{G}|)}{n}}\right).$$

Since we can write

$$\mathcal{F}_{N_T} = \{f_0\} + \sum_{i=1}^{N_T} (\mathcal{H}_1 + \mathcal{H}_2 + \mathcal{H}_3 + \mathcal{H}_4),$$

and the Rademacher complexity of a singleton set is 0, we can repeatedly apply Lemma F.4 to obtain

$$R_n(\mathcal{F}_{N_T}) \le N_T \cdot O\left(\sqrt{\frac{\log(|\mathcal{G}|)}{n}}\right) = O\left(N_T \sqrt{\frac{\log(|\mathcal{G}|)}{n}}\right).$$

This completes the proof. $\square$

### F.3 Optimizing over $\mathcal{F}_{N_T}$ and its Convergence Rate

As $N_T$ grows larger, more trees are added, which enables a more accurate optimization of the loss. We calculate the convergence of the tree ensemble as $N_T$ increases by analyzing the convergence rate of the optimization algorithm used to construct the tree ensemble.

Although LightGBM is often a practical choice for minimizing the loss and is used for our experiments, we do not restrict ourselves to any solver in Algorithm 1 in Section 4. For the purpose of this convergence analysis, we will analyze an alternative algorithm, SquareLev.R, introduced by Duffy and Helmbold Duffy & Helmbold (2002). All results and algorithm details in this subsection are derived from their work.

Like LightGBM, the SquareLev.R algorithm is a boosting algorithm designed to iteratively reduce the squared loss. It achieves this by minimizing the variance of the residuals at each step. The residual for sample $x_i$ at the beginning of step $k$ is defined as

$$r_i^{(k-1)} = y_i - F_{k-1}(x_i),$$

where $F_{k-1}(x_i)$ is the prediction of the ensemble after $k-1$ base learners have been added. The minimization of the variance of the residuals aligns with the purpose of loss minimization, as they are equivalent up to a constant shift of the predictor.

---

**Algorithm 2** SquareLev.R Algorithm Duffy & Helmbold (2002)

1: **Input:** A sample $S = \{(x_1, y_1), (x_2, y_2), \ldots, (x_m, y_m)\}$, a base learning algorithm, and parameters $\rho, T_{max}$
2: Set $D(x_i) \leftarrow 1/m$ and $t \leftarrow 1$
3: Initialize master function $F_1$ to the zero function
4: **for** $i = 1$ to $m$ **do**
5:      $r_i \leftarrow y_i - F_1(x_i)$                                ▷ Initial residuals, effectively $\mathbf{r}^{(0)}$
6: **end for**
7: **while** $\|r - \bar{r}\|_2^2 \geq m\rho$ and $t < T_{max}$ **do**
8:      $t \leftarrow t + 1$
9:      **for** $i = 1$ to $m$ **do**
10:          $\tilde{y}_i \leftarrow r_i - \frac{1}{m} \sum_j r_j$
11:      **end for**
12:      $S' \leftarrow \{(x_1, \tilde{y}_1), \ldots, (x_m, \tilde{y}_m)\}$
13:      Call base learner with distribution $D$ over $S'$, obtaining hypothesis $f$
14:      $\epsilon_t \leftarrow \frac{((r - \bar{r}) \cdot (f - \bar{f}))}{\|r - \bar{r}\|_2 \|f - \bar{f}\|_2}$
15:      $\alpha_t \leftarrow \frac{\epsilon_t \|r - \bar{r}\|_2}{\|f - \bar{f}\|_2}$
16:      $F_t \leftarrow F_{t-1} + \alpha_t f$
17:      **for** $i = 1$ to $m$ **do**
18:          $r_i \leftarrow y_i - F_t(x_i)$
19:      **end for**
20: **end while**
21: **Output:** $F_t$

---

The pseudocode for the SquareLev.R algorithm is presented in Algorithm 2. Based on Theorem 4.1 and 4.2 from Duffy & Helmbold (2002), we can characterize the convergence of the SquareLev.R algorithm using the Pearson correlation (also referred to as the "edge" in weak learning) between the residuals and the base learner output, specifically the $\epsilon_t$ term.

**Proposition F.9.** *The variance of the residuals is reduced by a factor of $(1 - \epsilon_k^2)$ during step $k$ in the SquareLev.R algorithm. If the edges of the weak hypotheses used by SquareLev.R are bounded below by $\epsilon_{min} > 0$, then the variance of residuals after $N_T$ steps, then the variance of residuals is bounded by*

$$Var(\mathbf{r}^{(N_T)}) \leq Var(\mathbf{r}^{(0)})(1 - \epsilon_{min}^2)^{N_T}.$$

Since the squared error of a shifted ensemble is equivalent to this variance of residuals, this implies a similar exponential reduction in the empirical loss.

## F.4 Combined Analysis and Sample Complexity

We now combine the Rademacher complexity analysis with the convergence rate results to establish the sample complexity for achieving a target multi-calibration error.

**Theorem F.10.** *With $n$ samples taken i.i.d. from the distribution $\mathcal{D}$, the excess risk of minimizing the loss on the ensemble $\mathcal{S}_{N_T}$ by running the SquareLev.R algorithm for $N_T$ iterations is bounded with probability $1 - \delta$ by*

$$O\left((1 - \epsilon_{min}^2)^{N_T}\right) + O\left(N_T \sqrt{\frac{\log(|\mathcal{G}|)}{n}}\right) + O\left(\sqrt{\frac{\log(1/\delta)}{n}}\right).$$

This result follows from combining the optimization error (governed by the convergence rate of the empirical loss) and the generalization error (governed by the Rademacher complexity).

Given that the multi-calibration error is bounded by the square root of the true expected loss $\epsilon_{loss}$ (as stated in Theorem 4.6), to achieve a multi-calibration error bounded by $\alpha$, we require that this true expected loss be bounded by $\alpha^2$. This leads to the following corollary

**Corollary F.11.** *If the number of trees $N_T$ and the sample size $n$ should satisfy*

$$N_T = O\left(\epsilon_{min}^{-2} \log(1/\alpha)\right)$$

*and*

$$n = \Omega\left(\alpha^{-4}\epsilon_{min}^{-4} \log(|\mathcal{G}|) \log^2(1/\alpha) + \alpha^{-4} \log(1/\delta)\right),$$

*minimizing the loss on ensembles defined in Equation (5) by running the SquareLev.R algorithm for $N_T$ iterations using $n$ samples taken i.i.d. from the distribution $\mathcal{D}$ gives a multi-calibration error less than $\alpha$ with probability $1 - \delta$ when the discretization error is small and Assumption 4.5 holds.*

*Proof.* For the multi-calibration error to be less than $\alpha$, we need the empirical risk, as bounded by Theorem F.10, to be $\leq \alpha^2$. We aim for each of the three terms in the bound to be $O(\alpha^2)$. For simplicity in balancing, let's set each term to be $\leq K\alpha^2/3$ for some constant $K$.

For the first term (empirical optimization error)

$$(1 - \epsilon_{min}^2)^{N_T} \leq \frac{K\alpha^2}{3},$$

taking logarithms and using the approximation $\log(1 - x) \leq -x$ for $x \geq 0$ gives

$$N_T \geq \frac{2\log(1/\alpha) - \log(K/3)}{\epsilon_{min}^2} = O\left(\epsilon_{min}^{-2} \log(1/\alpha)\right)$$

For the second term

$$N_T \sqrt{\frac{\log(|\mathcal{G}|)}{n}} \leq \frac{K\alpha^2}{3},$$

substituting $N_T = O(\epsilon_{min}^{-2} \log(1/\alpha))$ gives

$$n = \Omega\left(\alpha^{-4}\epsilon_{min}^{-4} \log^2(1/\alpha) \log(|\mathcal{G}|)\right).$$

And for the third term

$$\sqrt{\frac{\log(1/\delta)}{n}} \leq \frac{K\alpha^2}{3},$$

solving for $n$ gives

$$n = \Omega\left(\alpha^{-4} \log(1/\delta)\right).$$

Combining these constraints on $n$, the overall sample size is determined by the sum of these requirements

$$n = \Omega\left(\alpha^{-4}\epsilon_{min}^{-4} \log(|\mathcal{G}|) \log^2(1/\alpha) + \alpha^{-4} \log(1/\delta)\right).$$

This completes the proof. □

Corollary F.11 provides clear guidance on the relationship between the desired multi-calibration error $\alpha$, the number of trees $N_T$, and the required sample size $n$ to ensure the true expected loss is sufficiently small. This indicates that, even when we only have finite samples, the main conclusion still holds.

