# OpenReview forum: "Discretization-free Multicalibration through Loss Minimization over Tree Ensembles"
_NeurIPS.cc/2025/Conference — NeurIPS 2025 poster_

### Official Review · Reviewer_kSyY · 2025-06-24

**Clarity:** 2
**Significance:** 4
**Originality:** 3
**Rating:** 4
**Confidence:** 3

**Summary:**

The paper presents a novel multi-calibration method: given a base predictor $f_0$, the paper proposed to optimize the MSE by adding a series of depth-2 trees. A single depth-2 tree would partition the dataset into four subgroups, based on 1) whether $f_0(x) >= v$ and 2) whether the instances belong to a certain pre-defined group $i$.

Theoretical results show that such a algorithm "automatically" guarantees bounded multi-calibration error. Thus, the method does not depend on how we discretize the range of $Y$; although this is still needed to evaluate the multi-calibration error afterwards. Emprical experiments also domenstrate that the multi-calibration errors are lower than competitor methods.

**Questions:**

Besides my comments above, I have the following questions:

1) why using linear models for the tabular data in the experiment? This is related to my last point above: what would happen if you use, e.g., lightGBM, or another tree-boosting method like XGboost here?

2) Why some of the points are missing in Figure 2? Is it because of the too long runtime or something else?

3) I would appreciate an empirical verification of Thm 4.6, to further justify the assumptions used for proof. That is, to empirically compare the multi-calibration error with the $sqrt(\epsilon_{round} + \epsilon_{loss})$.

**Ethical Concerns:**

["NO or VERY MINOR ethics concerns only"]

**Final Justification:**

All my major concerns are addressed. The only minor issues left are about

1) the "overshooting" in the empirical validation of the theorem.
2) the presentation of the method in the figure, but this is minor (though it might cause some big confusions so please revise it).

**Limitations:**

Yes

**Quality:**

2

**Strengths And Weaknesses:**

I like the proposed direction in the paper; however I don't think the paper is mature enough to be published at a venue like NeurIPS.

Strenghs:
- The paper is well-structured. The story line is in general clear.
- The report empirical experiment results look quite strong.
- The paper takes a quite novel approach to tackle this problem (to the best of my knowledge).

However, I do have the following concerns:
1. In the descriptions of the technical part in Section 4, some parts are contradictory to each other.
- In eq.3, it does not make sense for $S_1(f_0)$ and $S_2(\mathcal{G})$ to take the union among all possible $v$ and $i$ respectively. I assume you want to use them to denote the collection of the subsets, not the union of the subsets?
- Eq.4 is contradictory to the illustration in Fig.1: in eq.4, $s_2$ is any FIXED element from $S_2(G)$, yet in Fig.1, it seems that $s_2$ can be different (you can take g_1(.) and g_2(.) for different leaves. )

2. The source code is not shared making the reproducibility of the results quite low. Any specific reason why the code is not shared to the reviewers?

3. It seems in the paper that the method can work with any model $f_0$; however, I am a bit skeptical in whether the proposed method would encounter a pitfall: e.g., when $f_0$ is trained with a tree boosting method (which is quite common for tabular data), and e.q.(5) is optimized with a tree boosting as well, then will the proposed method still learn to multi-calibrate? I would appreciate if this is at least discussed.

---

> ### Author Rebuttal · Authors · 2025-07-29
>
> We thank the reviewer for the careful review. We’re happy to respond to the questions raised by the reviewer.
>
> 1. Notation error in eq 3.
>
>     Indeed, according to eq 4, $s_1$ and $s_2$ should be sets, so $\mathcal{S}$ should be a collection of sets. The correct version of Eq. 3 should be $\mathcal{S}_1(f_0) = \\{ \\{ x \in \mathcal{X} : f_0(x) \geq v \\} : v \in R(f_0) \cup \\{0\\} \\} $ and $\mathcal{S}_2(G) = \\{ \\{ x \in \mathcal{X} : g_i(x) = 1 \\} : i \in [|G|] \\} .$ Thank you for noting this technical inaccuracy, and we’ll correct it in our manuscript.
>
> 2. Difference between Fig. 1 and Eq. 4
>
>     The two representations are different but theoretically equivalent. On one hand, the formulation in Eq. 4 can be represented by the one in Fig. 1 simply by setting the groups to be the same in a tree. On the other hand, a tree that has two different groups as features can be written as the sum of two trees in Eq. 4, one with $c_1=c_2=0$ and the other with $c_3=c_4=0$. In the illustration, we used different groups in the first place, as that’s what is implemented by the solver. We intend to change this to avoid confusion.
>
> 3. Source code availability
>
>     We have already made the codebase available online. Thank you for your interest in our implementation.
>
> 4. On the model architecture of the baseline algorithm for tabular data
>      Linear models are commonly used for tabular data in fairness literature, as evidenced by [1,2,3]. However, we recognize tree-boosting methods are also prevalent. In our derivation, we didn’t restrict the architecture of the uncalibrated baseline. Also, our method operates on different input features, and therefore does not contradict the baseline architecture.
>
>     We further conducted an experiment here in which tree-boosting architecture is used for the uncalibrated baseline:
>
>     - Experiment setting: instead of linear models, we use `LGBMClassifier(num_leaves=8)` and `LGBMRegressor(num_leaves=8)` for the classification and regression tasks respectively. The other hyperparameters are chosen by default.
>
>     - Results: We present the loss in Table 3 (complementing Table 1 in the paper) and worst-group smECE in Table 4 (complementing Table 2 in the paper).
>
>         Table 3: The empirical marginal improvement of running Algorithm 1 the second time.
>
>         | Task | $\ell(f_0)$ | $\ell(f^{cal})$ | $\ell(p_G(f^{cal}))$ | $\hat{\epsilon}_{loss} = \ell(f^{cal}) - \ell(p_G(f^{cal}))$ |
>         |------|-------------|-----------------|---------------------|---------------------------------------------------|
>         | Income Classification | 0.1285 | 0.1277 | 0.1276 | $2 \times 10^{-5}$ |
>         | Income Regression  | 0.0325 | 0.0322 | 0.0322 | $9 \times 10^{-7}$ |
>         | Travel Time Regression  | 0.0276 | 0.0276 | 0.0276 | $-3 \times 10^{-6}$ |
>
>         Table 4: The worst-group smECE of the evaluated algorithms. The values have been multiplied by 1000 for readability.
>
>         | Method                 | Discr.   | Income Class.   | Income Reg.   | Travel Time   |
>         |:-----------------------|:---------|:----------------|:--------------|:--------------|
>         | Uncalibrated Baseline  | /        | 93.20           | 63.27         | 24.93         |
>         | Multiaccurate Baseline | /        | 67.50 ± 12.76   | 41.13 ± 1.92  | 24.88 ± 2.23  |
>         | LSBoost                | 10       | 91.06 ± 3.29    | 50.56 ± 7.27  | 42.86 ± 0.07  |
>         |                 | 20       | 93.48 ± 1.22    | 51.16 ± 7.76  | 26.56 ± 0.22  |
>         |                 | 30       | 86.89 ± 6.18    | 50.78 ± 5.05  | 28.04 ± 0.75  |
>         |                 | 50       | 78.52 ± 8.01    | 56.51 ± 4.21  | 27.26 ± 2.65  |
>         |                 | 75       | 84.49 ± 7.46    | 56.99 ± 3.53  | 26.36 ± 1.50  |
>         |                 | 100      | 87.58 ± 4.21    | 57.69 ± 3.19  | 28.97 ± 3.17  |
>         | MCBoost                | 10       | 94.57 ± 0.00    | 62.99 ± 0.00  | 42.84 ± 0.00  |
>         |                 | 20       | 90.41 ± 3.21    | 64.95 ± 0.45  | 26.48 ± 0.00  |
>         |                 | 30       | 89.62 ± 2.49    | 63.78 ± 0.51  | 27.82 ± 0.71  |
>         |                 | 50       | 91.54 ± 2.02    | 62.26 ± 0.49  | 28.62 ± 1.89  |
>         |                 | 75       | 92.33 ± 1.82    | 63.06 ± 0.50  | 25.67 ± 1.39  |
>         |                 | 100      | 93.64 ± 1.62    | 62.70 ± 1.01  | 27.13 ± 0.36  |
>         | Ours                   | /        | **64.08** ± 13.65   | **38.34** ± 2.33  | **24.75** ± 2.20  |
>
>
>     Table 3 shows that Assumption 1 is still empirically valid in the new setting. Table 4 shows that using a tree-boosting baseline also produces remarkable multicalibration, even surpassing what we originally reported with linear base models. Therefore, our algorithm still works when the uncalibrated model is a tree ensemble.
>
> 5. Clarification of Fig. 2:
>
>     During discretization, we set the size of the codomain to be 10, 20, 30, 50, 75, 100, and there are always six points on each line in Fig. 2. From this perspective, no point is missing. We think what the reviewer meant by missing is that some lines don’t extend to 100. This is because the x axis in Fig. 2 is the size of the range, which can be smaller than the size of the codomain.
>
> 6. Empirical evaluation of the theorem
>
>     First, we would like to point out that it is not possible to directly evaluate $\epsilon_{loss}$, as discussed in 5.3.1. We use $\hat \epsilon_{loss}$, the empirical loss improvement instead as a proxy, with $\epsilon_{loss}=\hat \epsilon_{loss} + \epsilon_{opt} \geq \hat \epsilon_{loss}$, where $\epsilon_{opt}$ is the optimization error.
>
>
>     Denoting the multicalibration error as $\alpha$, we show that, empirically, $\alpha < \sqrt{\hat \epsilon_{loss} + \epsilon_{round}} \le \sqrt{\epsilon_{loss} + \epsilon_{round}}$ in most cases. We calculate $\alpha - \sqrt{\hat \epsilon_{loss} + \epsilon_{round}}$ and present it in the following table.
>
>     Table: $\alpha - \sqrt{\hat \epsilon_{loss} + \epsilon_{round}}$ of our algorithm on different tasks and different discretization, multiplied by 1000 for readability.
>     | Discretization | Income Class.   | Comment Toxicity Class.   | Age Reg.      | Income Reg.   | Travel Time Reg.   |
>     |--------------:|:----------------|:--------------------------|:--------------|:--------------|:-------------------|
>     |            10 | -19.69 ± 1.14   | -35.78 ± 0.76             | -19.15 ± 0.37 | -25.30 ± 0.87 | -17.55 ± 0.72      |
>     |            20 | -6.63 ± 0.98    | -16.86 ± 0.69             | -7.00 ± 0.70  | -11.93 ± 0.83 | -11.41 ± 0.62      |
>     |            30 | -2.22 ± 1.71    | -10.74 ± 1.07             | -4.84 ± 0.75  | -6.33 ± 0.90  | -7.86 ± 0.89       |
>     |            50 | 1.62 ± 1.58     | -5.75 ± 1.66              | -2.66 ± 0.54  | -1.39 ± 1.33  | -2.49 ± 1.01       |
>     |            75 | 2.68 ± 1.70     | -3.75 ± 1.98              | -1.31 ± 0.81  | 0.72 ± 1.62   | -0.63 ± 0.62       |
>     |           100 | 5.34 ± 2.02     | -2.03 ± 2.27              | -0.39 ± 0.75  | 2.11 ± 1.29   | 0.80 ± 0.61        |
>
>     Note that the task the task Skin Lesion Classification is not included here. This is because the empirical evaluation of $\hat \epsilon_{loss} + \epsilon_{round}$ is negative for this task.
>
>     As is shown here, we have $\alpha < \sqrt{\hat \epsilon_{loss} + \epsilon_{round}}$ in 80% of the different settings. Given $\epsilon_{loss}\geq \hat \epsilon_{loss}$, we can assert that $\sqrt{\epsilon_{loss} + \epsilon_{round}}$ is empirically larger than the multicalibration error in most cases, thus validating the theorem empirically.
>
> We hope this helps answer your questions.
>
> [1] Jung, Christopher et al. “Batch Multivalid Conformal Prediction.” International Conference on Learning Representations(2023).
>
> [2] Xian, Ruicheng et al. “Fair and Optimal Classification via Post-Processing.” International Conference on Machine Learning (2022).
>
> [3] Chen, Yatong et al. “Fairness Transferability Subject to Bounded Distribution Shift.” Advances in Neural Information Processing Systems (2022)

---

> ### Comment · Reviewer_kSyY · 2025-08-04
>
> Thanks a lot for the comments and the additional experiments..
>
> A few follow-up questions:
> - you wrote (Line 112) "Each decision tree of depth 2 in the ensemble assigns a real value ci 112 : i ∈ [4] to each of its 4 leaves", and each "tree-illustrating subfigure" (like the one under the title Tree 1) in figure 4 should correpond to a single tree. Hence, in my opinion, g1 must be equal to g2.
> Could you explain what do you mean by, here I quote, "In the illustration, we used different groups in the first place, as that’s what is implemented by the solver"?
>
> - For the point 4 in your rebuttal letter, about the experiment, when you use lightGBM for $f_0$, did you use LightGBM or XGboost to optimzie Eq (5) in the paper? If you use the LightGBM, optimizing (5) is somewhat like adding more iterations in the boosting, with different attributes for the trees of course, but the attributes like $f_0 > 3$ in figure 1 can theoretically represented by sums of trees constructed by original features (i.e., the X's).. Thus, I find it hard to intuitively understand why it works. That's also why I want to check your implementation to see if something is missing in the descriptions.
>
> - Regarding the point 6 in your rebuttal letter, I agree that negative values mean good, but I also saw a disturbing trend that as the number of discretization increases, $\alpha - \sqrt{\hat \epsilon_{loss} + \epsilon_{round}}$ also increases. For 3 out 5 datasets you show, the increase does not stop at 0 but seem to overshoot and keep increasing to larger positive numbers... Have you noticed this and it there any explanation?

---

> > ### Author Response · Authors · 2025-08-05
> > **Responses to the follow-up**
> >
> > - On the clarification of our rebuttal point 2
> >
> >     We are aware that, according to line 112 and Eq. 4, we require that each tree should only address a single group; yet our graph contradicts this. Since our graph is an illustration of our framework, we agree that it should be consistent with Eq. 4, and we’ll change it.
> >
> >     As for the quoted sentence, we were trying to explain why this contradiction was brought to our manuscript in the first place. Many tree ensemble solvers, like XGBoost and lightGBM, allow different features on the same level, and we mistakenly include this solver-specific feature into our illustration of our solver-independent framework. We did explain why these two formulations are equivalent in our previous response though.
> >
> >     Hope that this answers your confusion!
> >
> >  - On why thresholding on $f_0$ cannot be replaces by additional trees
> >
> >     We did use LightGBM for the base predictor, but it’s not simply equivalent to adding more iterations (or, more trees). The feature $f_0(x)$ that is exclusively for our algorithm is the key difference, even when $f_0$ itself is an ensemble of trees. We would like to point out that (a) learning an ensemble of depth-m trees and splitting on its outputs, is not equivalent to (b) learning an ensemble of depth-(m+1) trees. The former has a larger capacity than the latter.
> >
> >     Let’s illustrate this point by setting $m=1$. Say there are $n$ binary features. It’s possible for a ensemble of $n$ depth-1 tree to output $2^n$ different outputs, and with an additional ensemble of depth-1 trees that take the thresholds on such outputs, it is theoretically possible for (a) to assign different values to all $2^n$ configurations of the $n$ binary features. However, the number of unique trees in ensemble (b) is at most polynomial in $n$.
> >
> >     Of course in practice, to ensure generalization, the output $f_0(x)$ is treated as a single continuous feature instead of an indicator for every single datapoint; but we hope that through this example, you can see that even in the specific case where tabular data is already fitted with an ensemble and where the group indicators can be represented by the original features, our post-processing algorithm still cannot be absorbed into $f_0$ and be written as a larger ensemble with extended iterations.
> >
> >  - On the explanation of the empirical results in the additional experiment
> >
> >     Building on our previous discussion, the empirical result serves as a loose upper bound, which explains the positive values we observe. As for the trend, while it’s hard to provide a formal analysis, we can try to understand it intuitively. Let’s assume that the discretizing $[0,1]$ to $m$ level sets adds a perturbation $\delta$ to the output $f(x)$, where $\delta$ is a uniform random variable in $[-1/2m, 1/2m]$. If the mean of the predictor output $E[f(x)]$ is the same as the mean of the ground truths $E[y]$, then the discretization error is $E[(f(x)+\delta-y)^2-(f(x)-y)^2]=E[\delta^2]=\frac{1}{12m^2}$, which increases fast as $m$ increase. However, the multicalibration error remains the same as $E[\delta]=0$. Then, the subtracted value will become more negative as $\delta$ increases, namely as $m$ decreases.
> >
> > We appreciate your feedback, and hope this discussion addresses your concerns!

---

> > > ### Comment · Reviewer_kSyY · 2025-08-06
> > >
> > > Thanks for the further feedback. Regarding your three points:
> > >
> > > - I don't agree it is equivalent. As said, it is only equivalent if g1 is equal to g2.
> > >
> > > - Please do notice that in my previous reply, I was aware that using $f_0 > 3$ as the splitting condition in the tree is not the same as using original features. Your reply hence does not answer my question. My question was about that, since $f_0$ is a tree ensemble as well, the condition $f_0 > 3$ should be able to be represented by the sum of the trees (precisely leaves, actually). In that sense, it is hard to intuitively understand whether you could obtain "new information" to further decrease the loss. I'd appreciate if you could elaborate here, but I do understand that it might work in practice indeed, as shown by your experiments.
> > >
> > > - I think your reply contains errors, which make it hard to follow. For instance, $E[(f(x)+\delta-y)^2-(f(x)-y)^2]=E[\delta^2]=\frac{1}{12m^2}$, which increases fast as $m$ increase. I believe as $m$ increases $\frac{1}{12m^2}$ will only decrease.
> > > Further, I don't see why "loose upper bound" can explain "positive results" here.

---

> ### Author Response · Authors · 2025-08-08
> **Further explanation (1/3)**
>
> Thank you for your continued interest in the details of our paper and for your patience with us to discuss them.
>
> The difference between the ensembles is a very important question on the ensemble of decision trees that our proposed algorithm optimizes over, which is $\{\sum_{t \in T }t : T\subseteq \mathcal{T}(f_0,\mathcal{G}) \}$ in Equation (5) of our paper.
>
> In the theoretical analysis of the paper, for simplicity, we define $\mathcal{T}(f_0,\mathcal{G})$ as in Equation (4), hereafter we call this $\mathcal{T_1}(f_0,\mathcal{G})$. In practice, many tree ensemble solvers, like XGBoost and lightGBM, allow different features on the same level. So the actual decision trees to be ensembled are in the form illustrated in Figure 1, where $g_1$ and $g_2$ are not necessarily the same. We call this class of decision trees $\mathcal{T_2}(f_0,\mathcal{G})$:
> $$ \mathcal{T_2}(f_0,\mathcal{G})=  \\{c_1\cdot I\\{x\in s_1 \wedge x\in s_2\\}+c_2\cdot I\\{x\in s_1 \wedge x\notin s_2\\}+ c_3\cdot I\\{x\notin s_1 \wedge x\in s_3\\} +c_4\cdot I\\{x\notin s_1 \wedge x\notin s_3\\}   :c_1,c_2,c_3,c_4\in  \mathbb{R}, s_1\in  \mathcal{S}_1(f_0),s_2, s_3\in  \mathcal{S}_2(\mathcal{G})\\}. $$
>
> Below we show     $$ \\{\sum_{t \in T }t : T\subseteq \mathcal{T_1}(f_0,\mathcal{G}) \\} = \\{\sum_{t \in T }t : T\subseteq \mathcal{T}_2(f_0,\mathcal{G}) \\} .$$
>
>
> It is obvious that $\mathcal{T_1}(f_0,\mathcal{G}) \subseteq \mathcal{T_2}(f_0,\mathcal{G})$. Therefore $\\{\sum_{t \in T }t : T\subseteq \mathcal{T_1}(f_0,\mathcal{G}) \\} \subseteq \\{\sum_{t \in T }t : T\subseteq \mathcal{T_2}(f_0,\mathcal{G}) \\}$.
>
> On the other hand, for any $t\in  \mathcal{T_2}(f_0,\mathcal{G})$, it can be written as the sum of two trees in $\mathcal{T_1}(f_0,\mathcal{G})$. One tree has $g_1$ on the second level and $c_3 = c_4 = 0$. The other tree has $g_2$ on the second level and $c_1 = c_2 = 0$.
>
> More concretely, let $t = c_1\cdot I\\{x\in s_1 \wedge x\in s_2\\}+c_2\cdot I\\{x\in s_1 \wedge x\notin s_2\\}+ c_3\cdot I\\{x\notin s_1 \wedge x\in s_3\\} +c_4\cdot I\\{x\notin s_1 \wedge x\notin s_3\\}$, where $c_1,c_2,c_3,c_4\in  \mathbb{R}, s_1\in  \mathcal{S}_1(f_0),s_2, s_3\in  \mathcal{S}_2(\mathcal{G})$. $t = t_1 + t_2$, where $t_1 = c_1\cdot I\\{x\in s_1 \wedge x\in s_2\\}+c_2\cdot I\\{x\in s_1 \wedge x\notin s_2\\}+ 0\cdot I\\{x\notin s_1 \wedge x\in s_2\\} +0\cdot I\\{x\notin s_1 \wedge x\notin s_2\\}$ and $t_2 = 0\cdot I\\{x\in s_1 \wedge x\in s_3\\}+0\cdot I\\{x\in s_1 \wedge x\notin s_3\\}+ c_3\cdot I\\{x\notin s_1 \wedge x\in s_3\\} +c_4\cdot I\\{x\notin s_1 \wedge x\notin s_3\\}$. $t_1, t_2 \in  \mathcal{T_1}(f_0,\mathcal{G})$.
>
> Therefore, $\\{\sum_{t \in T }t : T\subseteq \mathcal{T_1}(f_0,\mathcal{G}) \\} =\\{\sum_{t \in T }t : T\subseteq \mathcal{T_2}(f_0,\mathcal{G}) \\}$. The two ensembles are equivalent.

---

> ### Author Response · Authors · 2025-08-08
> **Further explanation (2/3)**
>
> Thank you for the letting us know that we agree that splitting condition in the tree is not the same as using original features.
> So you're asking: since $f_0$ is itself a tree ensemble:
> $$f_0(x) = T_1(x) + T_2(x) + \ldots + T_M(x),$$
> shouldn't the threshold condition, like $f_0(x) > 3$, be representable through some combination of the existing leaves in those trees? And if so, why does our post-processing provide any benefit—what "new information" are we gaining?
>
> -  First, we'd like to point out that for decision trees ensembles, like LightGBM and XGBoost, all the trees use the same set of features that are fixed at the beginning of the algorithm; usually it does not allow taking the intermediate outputs, like the constructed leaves, or the output of the trees constructed so far, as the feature. Therefore, although $f_0$ can be represented by the sum of the trees, taking the threshold on $f_0$ is not a valid operation in tree ensembles. We believe this might be the major source of confusion, because this is actually different from the notion of boosting as in the paper [1], where the output of the model in the current step can be used as the input feature of the next timestep. Therefore, our method is very different from adding more iterations in the boosting.
>
>  - Second, we'd like to explain why the loss can be further decreased.
> You're correct that no new information about the underlying data distribution is gained, and that threshold conditions can theoretically be derived using the original features, though not with standard tree ensembles only.
> The key insight is, instead of gaining new information, we are gaining access to decision boundaries that lie outside the representational capacity of standard tree ensembles. More precisely, we'll show that: given $n$ binary features $x$, there exists an ensemble $f_0$ of trees with depth $d_0$ such that representing all threshold functions on $f_0$ requires trees of depth $n$.
>
>     - A simple example to see why this is true is to set $f_0=\sum_{i=1}^n 2^{-i} x_i$. Then, $f_0$ is an one-to-one function that has $2^n$ discrete outputs. With two thresholding functions $I\\{f(x)\ge c\\}$ and $I\\{f(x)\ge c+2^{-n}\\}$, we can obtain any indicator $I\\{x=x_0\\}$ for any $x_0$, with
>         $$I\\{x=x_0\\}=\prod_{j=1}^{n}\left((x_j)^{(x_0)_j}(1-x_j)^{1-(x_0)_j}\right),$$
>         where we assume $0^0=1$.
>
>         Now ensembles of depth-$d$ trees can be written as a polynomial of degree $d$:
>
>         $$\sum_{l} v_l \prod_{j \in  T_l} x_j^{b_{l,j}}(1-x_j)^{1-b_{l,j}}$$
>
>         where $T_l$ contains the indices $j$ of features $x_j$ that are split on in the path to leaf $l$, $|T_l| \le  d$ is the depth constraint, and $b_{l,j} \in  \\{0,1\\}$ indicates whether feature $x_j$ appears in its original form ($b_{l,j}=1$) or negated form ($b_{l,j}=0$) in the leaf condition.
>
>         If two polynomials of degree $d$ subtracts to an irreducible polynomial of degree $n$, the only possibility is that $d\ge n$. Since we only have $n$ binary features, we got $d=n$. In fact, this argument holds as long as $f_0$ is one-to-one.
>
>     That is to say, if we want all threshold functions to be represented by an ensemble of trees as well, this ensemble has to consist of depth-$n$ trees. Indeed, if we can learn an $f_0$ that consists of trees of depth $d_0=n$, there will not be any improvement on the representation power by post-processing, and indeed our algorithm will not be beneficial.
>
>     And for the general case where $d_0<n$, thresholding functions on $f_0$, which is an ensemble of trees of depth $d_0$, cannot be written as an ensemble of depth-$d$ trees for any $d<n$ in general. This is where the gain in loss comes from, as we learn the same data with a more powerful model.
>
>  - Third, the model could also benefit from feature engineering even when no new information is present. We'd like to further illustrate the effect of the groups here. Even the group indicators are obtainable from the original features, explicitly including them in the input of our algorithm serves the role of feature engineering. This doesn't introduce new information neither, but it makes the model more focused on the multicalibration task, which also contributes to the decreased multicalibration error.
>
> [1] Ira Globus-Harris, Declan Harrison, Michael Kearns, Aaron Roth, and Jessica Sorrell. 2023. Multicalibration as boosting for regression. In Proceedings of the 40th International Conference on Machine Learning (ICML'23), Vol. 202. JMLR.org, Article 460, 11459–11492.

---

> ### Author Response · Authors · 2025-08-08
> **Further explanation (3/3)**
>
> We agree that there were some wording issues in our previous response for point 3, so we’d like to explain it again in a more straightforward way.
>
> First, it’s hard to formally analyze this trend, so we try to intuitively understand why $\alpha-\sqrt{\epsilon_{loss}+\epsilon_{round}}$ is increasing as $m$ increases.
>  - $\epsilon_{loss}$, as defined in Assumption 4.5, is the additional loss decrease by optimizing Eq. 5 a second time, and this does not change with discretization.
>  - For an intuitive understanding, we consider the simplified case where the predictions are uniformly distributed in [0,1]. Then, discretizing to $m$ outputs can be viewed as adding a perturbation $\delta$ to the predictions, with $\delta$ being a uniform random variable in $[-1/2m, 1/2m]$. For the discretization error,
>  $$\epsilon_{round}=E[(f(x)+\delta-y)^2-(f(x)-y)^2]=E[\delta^2]=\frac{1}{12m^2}$$
> **decreases** fast as $m$ increases (previously a typo here).
>  - On the other hand, the multicalibration error is calculated using sums of $\left| E[f(x)]-E[y] \mid f(x), g(x) \right\|]$ where the expectation is taken within the absolute values. Intuitively, since $E[f(x)+\delta]=E[f(x)]$, $\alpha$ can remain unchanged as $m$ changes.
>
> This constitutes an intuitive understanding of why $\alpha-\sqrt{\epsilon_{loss}+\epsilon_{round}}$ increases as $m$ increases.
>
> Moreover, we reported $\alpha-\sqrt{\hat \epsilon_{loss}+\epsilon_{round}}$, which is an empirical upper bound of $\alpha-\sqrt{ \epsilon_{loss}+\epsilon_{round}}$, as we leave out the optimization error $\epsilon_{opt}$ in our evaluation. We know for sure that $\alpha-\sqrt{ \epsilon_{loss}+\epsilon_{round}}$ will be smaller than the reported values, but unfortunately, we do not have enough tools at this moment to quantify such differences.
>
> We hope that this explanation could be of help!

---

> > ### Comment · Reviewer_kSyY · 2025-08-08
> >
> > Hi,
> >
> > Thanks a lot for the very detailed explanations.
> >
> > I think if including $f_0$ as new features can further optimize the loss is true in general for tree boostings, this observation deserves some more discussion. Of course for this paper, the tabular data is only part of the experiments, hence not super important for now.
> >
> > I am still a bit concerned about the overshooting, but I am willing to consider this as a issue for future work, as the contribution now seems enough.
> >
> > I will increase my score accordingly.

---

### Official Review · Reviewer_zBZu · 2025-06-28

**Clarity:** 2
**Significance:** 2
**Originality:** 2
**Rating:** 4
**Confidence:** 2

**Summary:**

This paper studies the problem of multicalibration, where a predictor is calibrated for a large family of groups. It proposes a one-shot, discretization-free method, which treats the uncalibrated score as another feature, fits an L2-boosted forest so that once it cannot lower the loss further, the model is multicalibrated across all groups without choosing bins by hand. The paper provides theoretical support and experiments across six datasets, achieving the lowest worst-group Smooth-ECE on each dataset.

**Questions:**

See weaknesses.

**Ethical Concerns:**

["NO or VERY MINOR ethics concerns only"]

**Limitations:**

yes

**Quality:**

2

**Strengths And Weaknesses:**

Strengths:
- The paper seems to provide a nice theoretical link between loss and multicalibration.
- The results show that the method improves worst-group calibration.

Weaknesses:
- Does the algorithm affect accuracy of the predictors? I may have missed this but it looks like only calibration error is reported.
- How scalable is this method to many groups? Can this approach extend to implicit groups defined by unlabeled features?
- Does the method rely on square loss? Would it work to minimize cross-entropy instead?
- Can you include a baseline that is not multicalibration-aware just to see how much calibration error remains?
-  The method requires explicit group features, which may be difficult to obtain.

---

> ### Author Rebuttal · Authors · 2025-07-30
>
> We thank the reviewer for the careful review, and we’re happy to answer the questions raised.
>
> 1. On the accuracy of the predictors
>
>     This is a major aspect that practitioners might wonder about, and we thank the reviewer for pointing this out. Apart from calibration, other metrics might be of interest when evaluating an algorithm. Here we present the accuracy (for classification tasks) and MSE error (for regression tasks)
>
>     Table 1: The accuracy (%) of the predictors on classification tasks. The higher, the better.
>
>     | Method                 | Discr.   | Skin Lesion Class.   | Income Class.   | Comment Toxicity Class.   |
>     |:-----------------------|:---------|:---------------------|:----------------|:--------------------------|
>     | Uncalibrated Baseline  | /        | 70.21                | 76.71           | 92.88                     |
>     | Multiaccurate Baseline | /        | 73.19 ± 0.31         | 78.97 ± 0.06    | 92.89 ± 0.00              |
>     | LSBoost                | 10       | 77.22 ± 1.00         | 77.96 ± 0.25    | 92.81 ± 0.08              |
>     |                 | 20       | 76.55 ± 0.95         | 78.09 ± 0.33    | 92.78 ± 0.06              |
>     |                 | 30       | 76.86 ± 0.46         | 78.16 ± 0.21    | 92.78 ± 0.06              |
>     |                 | 50       | 77.06 ± 0.52         | 78.11 ± 0.20    | 92.71 ± 0.07              |
>     |                 | 75       | 76.52 ± 0.72         | 77.81 ± 0.27    | 92.65 ± 0.04              |
>     |                 | 100      | 76.12 ± 0.50         | 77.60 ± 0.30    | 92.81 ± 0.05              |
>     | MCBoost                | 10       | 71.00 ± 1.01         | 76.71 ± 0.00    | 92.88 ± 0.00              |
>     |                 | 20       | 72.15 ± 1.63         | 77.10 ± 0.34    | 92.88 ± 0.00              |
>     |                 | 30       | 73.54 ± 1.24         | 77.23 ± 0.20    | 92.88 ± 0.00              |
>     |                 | 50       | 72.07 ± 1.01         | 77.20 ± 0.40    | 92.88 ± 0.00              |
>     |                 | 75       | 73.12 ± 1.00         | 76.89 ± 0.17    | 92.88 ± 0.00              |
>     |                 | 100      | 72.29 ± 1.08         | 76.70 ± 0.22    | 92.87 ± 0.01              |
>     | Ours                   | /        | **78.90** ± 0.17         | **79.27** ± 0.07    | **92.93** ± 0.01              |
>
>     Table 2: The MSE loss ($\times 10^{-3}$) of the predictors on regression tasks. The lower, the better.
>
>     | Method                 | Discr.   | Age Reg.    | Income Reg.   | Travel Time Reg.   |
>     |:-----------------------|:---------|:------------|:--------------|:-------------------|
>     | Uncalibrated Baseline  | /        | 5.11        | 41.39         | 30.19              |
>     | Multiaccurate Baseline | /        | 3.51 ± 0.02 | 37.92 ± 0.02  | 29.76 ± 0.02       |
>     | LSBoost                | 10       | 2.01 ± 0.24 | 39.66 ± 0.35  | 31.06 ± 0.07       |
>     |                 | 20       | 1.42 ± 0.16 | 38.98 ± 0.31  | 30.29 ± 0.09       |
>     |                 | 30       | 1.36 ± 0.13 | 39.03 ± 0.23  | 30.25 ± 0.04       |
>     |                 | 50       | 1.52 ± 0.17 | 39.51 ± 0.22  | 30.16 ± 0.03       |
>     |                 | 75       | 1.53 ± 0.19 | 38.83 ± 0.19  | 30.13 ± 0.03       |
>     |                 | 100      | 1.72 ± 0.20 | 38.80 ± 0.20  | 30.17 ± 0.03       |
>     | MCBoost                | 10       | 5.08 ± 0.25 | 42.18 ± 0.01  | 31.23 ± 0.00       |
>     |                 | 20       | 4.69 ± 0.30 | 41.10 ± 0.23  | 30.42 ± 0.00       |
>     |                 | 30       | 4.92 ± 0.17 | 40.70 ± 0.26  | 30.31 ± 0.00       |
>     |                 | 50       | 4.40 ± 0.38 | 40.93 ± 0.27  | 30.22 ± 0.00       |
>     |                 | 75       | 3.81 ± 0.36 | 40.72 ± 0.32  | 30.22 ± 0.01       |
>     |                 | 100      | 3.88 ± 0.41 | 40.95 ± 0.29  | 30.18 ± 0.03       |
>     | Ours                   | /        | **1.08** ± 0.02 | **36.11** ± 0.11  | **29.80** ± 0.04       |
>
>     By optimizing for the loss directly, our algorithm performs slightly better on standard metrics (MSE error / accuracy) on all the tasks evaluated.
>
> 2. On the scalability w.r.t. number of groups.
>
>     Our approach scales well, which has been supported by both theoretical and empirical analysis. In theory, we selected a specific boosting algorithm that can be used in our framework and conducted finite-sample analysis in Appendix E. We showed that the sample complexity scales logarithmically with the number of groups. Empirically, we evaluated on approximately 50 groups for tabular data, substantially exceeding prior empirical multicalibration studies (e.g., Hansen et al. utilized at most 20 groups).
>
> 3. On the selection of loss function.
>
>     Unfortunately we didn’t support other loss functions for now. We acknowledge that cross entropy is more prevalent than Brier score loss for classification tasks, and this could be left for future investigations.
>
> 4. On baselines that are not multicalibration aware.
>
>     We did include non-multicalibration-aware baselines in our evaluation. The Multiaccurate Baseline achieves multiaccuracy (a weaker condition than multicalibration) and consistently exhibits higher calibration error than other algorithms.
>
> 5. On the requirement of explicit group features
>
>     Indeed, we require the group features to be explicitly given, and the model's performance on unseen groups is not guaranteed. This represents a limitation of our current approach. However, as established in the algorithmic fairness literature, many works in algorithmic fairness require that “variables are explicitly defined as ‘sensitive’ by specific legal frameworks”. [1] While extending to implicit group discovery constitutes important future work, explicit group definition remains the predominant paradigm in fairness-aware machine learning applications.
>
>     We would also like to add that our response point 3 to Reviewer 3k17 might be related, in which we discussed how certain group structures can be implicitly defined.
>
> [1] Simon Caton and Christian Haas. 2024. Fairness in Machine Learning: A Survey. ACM Comput. Surv. 56, 7, Article 166 (July 2024), 38 pages. https://doi.org/10.1145/3616865

---

### Official Review · Reviewer_Wact · 2025-06-28

**Clarity:** 3
**Significance:** 3
**Originality:** 3
**Rating:** 5
**Confidence:** 4

**Summary:**

The paper proposes a new multicalibration algorithm which does not require specifying the granularity of the level sets at training time. This is because the algorithm works by finding a subset of decision trees (on particular features such as group membership / prediction thresholds) which minimize the squared loss. This stands in contrast to most previous iterative-patching style algorithms.

The main theoretical result is a statement on the multicalibration error achieved by the post-processing algorithm. In particular, assuming that the post-processing algorithm does not improve the loss by much, and also that the discretization error is low, then the multicalibration error is also low.

Finally, experiments are conducted in comparing the proposed post-processing algorithm to LSBoost and MCBoost, two previous multicalibration algorithms.

**Questions:**

I had some questions on the evaluations run in the paper.
1. When reporting the results of the uncalibrated baseline, is the baseline predictor trained on the entire train + calib set, or just the train set? The reason I ask is because Hansen et al. argue that we should evaluate mcb post-processing algorithms w.r.t. the baseline which uses all available training data (since that is what a practitioner would probably try first).
2. Similarly, it seems that the baseline DistillBert model for CivilComments used in the submitted paper seems quite weak. In particular, Hansen et al. seem to use the same groups and model architecture, and achieve a much better baseline worst-group smECE of around 0.06 (compared to the 0.119 reported in this paper). Therefore, all multicalibration post-processing algorithms “appear” less useful, since the baseline is already so good. Improving the baseline model may decrease the gap / improvement between standard mcb post-processing methods and the proposed decision tree ensemble approach. (If you are struggling to improve the worst-group smECE of the Distilbert model, one tip is to leave it training for longer!)

I think that these are minor quibbles given that the proposed post-processing method seems to clearly improve on the tested baselines. In particular, the fact that one could get better baselines in certain regimes --- say, for the experiments with NNs --- does not detract from the overall results or message, which holds for all model families, including simple models on tabular baselines.

**Ethical Concerns:**

["NO or VERY MINOR ethics concerns only"]

**Final Justification:**

The authors answered the few questions I had. The work remains a strong contribution to the empirical multicalibration literature, and may eventually become the de-facto multicalibration algorithm for practitioners.

**Limitations:**

Yes

**Quality:**

4

**Strengths And Weaknesses:**

I previously served as an emergency reviewer for this paper (and hence, re-used the summary above).

In my opinion, the most important strength of the paper is that the multicalibration post-processing algorithm it proposes seems to be extremely effective (conditioned on the evaluations run by the paper).

At the time of my previous review, my main concerns / weaknesses were the following:

1. Motivation for focusing on the “discretization-free” approach of the paper.
2. Not evaluating worst-group calibration error within the experiments, which is a natural metric that practitioners may be interested in.

The first point (1) seems to be better addressed in this version of the paper. In particular, the second paragraph of the introduction motivates why one would want to get rid of the multicalibration discretization parameter with five concrete reasons:
- Introduces rounding error which leads to prediction distortion
- Adds a (sensitive) additional hyperparameter to model tuning
- The parameter introduces a bias / variance tradeoff
- Instability under different parameter choices
- Downstream decision makers with different utility functions may not get predictions with enough granularity in order to make a correct decision.

I think all of these are reasonable and effective motivations to get rid of the discretization parameter, and hence, I feel point (1) is no longer a major weakness of the paper. I would appreciate additional citations for the first and fifth bullets if possible!

The second weakness I had, point (2), also seems to have been addressed. In particular, the paper evaluates on the worst group smECE in Table 2, where they show that their proposed post-processing method outperforms all other baselines. In the prior version, the paper evaluated on the multicalibration error of a rounded predictor, which was a bit of an odd metric to evaluate on.

Since both of the major weaknesses were addressed, I am happy to advocate for accepting the paper.

---

> ### Author Rebuttal · Authors · 2025-07-28
>
> We thank the reviewer for the careful review. We are happy to respond to the questions raised by the reviewer.
>
> - On the dataset partitioning for the training of the baseline:
>
>   Indeed, our uncalibrated baseline models were trained only on the training set, not on the combined training + calibration data. While training a separate model with the combined dataset may decrease the performance gap between the uncalibrated models and the multicalibration ones, this won’t affect the gap among the multicalibration models, as they all post-processed the uncalibrated model (trained on only the training set) using a held-out calibration dataset.
>
>   The difference in this setting stems from the purpose of the experiments: Hansen et al. focused on whether multicalibration algorithms are necessary or not, i.e., if setting aside some data for different multicalibration algorithms will be beneficial compared to the baseline algorithms; while we focused on, given the same baseline algorithm and a calibration set, how well our algorithm perform compared to others. If considered this way, our setting should make sense as well.
>
> - On the discrepancy of the performance of the uncalibrated model on CivilComments dataset:
>
>   Thank you for pointing this out! Due to the aforementioned difference in data partitioning, we used 225,000 samples for baseline training, while Hansen et al. used 270,000 for the baseline algorithm. We also followed the suggested setting of finetuning for 5 epochs suggested by Koh et al. Indeed, it seems that training for additional 5 epochs might increase the performance of the uncalibrated predictor, as indicated by the results of Hansen et al., and we thank the reviewer for the kind tip.
>
> - Additional citations
>
>     As for the bibliographyic reference of the motivation, we find this work [1] relevant. Our motivation is quite similar to the motivating example of this paper, namely, down-stream decision-makers hope that the same model's predictions "lead to beneficial decisions according to their own loss functions".
>
>     Another work [2] might be relevant as well, as it discussed how information loss "caused by rounding can influence the predictive power of the machine learning models".
>
> [1] Zhao et al. (2021). Calibrating Predictions to Decisions: A Novel Approach to Multi-Class Calibration. NeurIPS 2021.
>
> [2] Senavirathne, N., Torra, V. Rounding based continuous data discretization for statistical disclosure control. J Ambient Intell Human Comput 14, 15139–15157 (2023)

---

> > ### Comment · Reviewer_Wact · 2025-08-04
> >
> > Thank you for the response! I will keep my score.

---

### Official Review · Reviewer_sZFF · 2025-06-29

**Clarity:** 4
**Significance:** 3
**Originality:** 3
**Rating:** 4
**Confidence:** 3

**Summary:**

The paper proposes a novel multicalibration algorithm that relaxes the discretization preprocessing typically required in existing multicalibration methods. The theoretical results hold for any Discretization Operation as defined by the authors. The core idea is to use depth-two decision trees trained via empirical risk minimization (ERM). Under the loss saturation assumption, the algorithm achieves multicalibration guarantees.

To demonstrate its practicality, the authors conduct empirical evaluations on multiple datasets, showing improved performance over existing multicalibration algorithms.

**Questions:**

In Figure 2, what exactly is meant by “different discretization”?  What is the significance of using different discretizations in this plot?

**Ethical Concerns:**

["NO or VERY MINOR ethics concerns only"]

**Final Justification:**

The authors have fully answered to my questions.

**Limitations:**

Yes

**Quality:**

3

**Strengths And Weaknesses:**

Strengths
1. The method generalizes previous approaches by relaxing the discretization requirement, enabling application to a broader class of discretization strategies.
2. The use of simple depth-two decision trees enhances both practical usability and interpretability.
3. The paper is clearly written and easy to follow.

Weaknesses
The loss saturation assumption is essential to the theoretical guarantee. While it often holds in practice, the authors provide a specific counterexample where it fails.

---

> ### Author Rebuttal · Authors · 2025-07-28
>
> We thank the reviewer for the careful review. We are happy to respond to the question raised by the reviewer.
>
> By discretization we mean restricting the codomain of the predictor to a discrete set, and a different discretization means a different codomain. For uncalibrated baseline, multiaccurate baseline and our algorithm, it's done by rounding the real-valued output to the nearest value in the codomain; and for LSBoost and MCBoost, it's done by setting the hyperparameter of the algorithm before running it.
>
> We vary the discretization for these two reasons:
>
> 1. Evaluation requirement: Multicalibration error, as defined in Definition 3.2, can only be computed for predictors with finite output ranges, so even our continuous predictor must be discretized for evaluation purposes. This metric can be sensitive to discretization, so we vary it for a more robust evaluation.
>
> 2. Practical significance: Different downstream applications may require different precision levels or thresholds for decision-making. Varying the discretization strengthens the point that a single trained predictor maintains low multicalibration error across various discretization schemes, making it more flexible for diverse applications without retraining.

---

> ### Comment · Reviewer_sZFF · 2025-08-06
>
> I appreciate the response from the authors.

---

### Official Review · Reviewer_3K17 · 2025-07-02

**Clarity:** 3
**Significance:** 3
**Originality:** 3
**Rating:** 5
**Confidence:** 3

**Summary:**

This paper tackles multicalibration without the discretization step required by prior methods. The authors proposed a simple one short learning algorithm that fits a LightGBM with depth 2, which effectively models the interaction of base predictor and the grouping indicators. They also proved that, if the learned ensemble is ”loss-saturated”,  then the multicalibration error is bounded. Empirically, across 6 datasets, the method has the lowest worst group smECE in general compared to baselines.

**Questions:**

Questions:

1. Could you comment on how the algorithm behaves when relevant sub-groups are not explicitly encoded? How sensitive is it to “hidden” groups that fall outside of the provided indicators?
2. Have you experimented with tree depth greater than 2? Could deeper trees help capture interactions among explicit groups and extend to a broader group family?
3. Are there any empirical details on the number of boosting rounds you used, as well as that used by LSBoost, at the same tree depth? I think comparing the number of round convergence of “one-shot” and “recursive” algorithms is quite interesting.

**Ethical Concerns:**

["NO or VERY MINOR ethics concerns only"]

**Final Justification:**

All the main concerns are solved. W1 is partially addressed and could be an interesting further direction.

**Limitations:**

Yes.

**Quality:**

3

**Strengths And Weaknesses:**

Strengths:

1. This paper developed a novel technique that got rid of the discretization step widely used by previous methods.
2. This paper used a single Ensemble tree fit to capture the interactions between the base prediction and group indicators, which is concise and easy to implement.
3. The empirical evaluation showed improved performance across 6 datasets over existing approaches on multicalibration error and worst group smECE.
4. The paper is well-written with theoretical justifications.

Weaknesses:

1. Unlike LSBoost that calibrates with respect to a function class, the method this paper introduced only guarantees the application to pre-specified groups. The method could suffer from unspecified important subgroups or high-dimension grouping structure.
2. Section 5.3.2 reports the discretized version of the proposed predictor, however, the author doesn’t show the undiscretized version, but only showed the worst group smECE in Section 5.3.3 instead. I understand that comparison in 5.3.2 is already fair, but adding the undiscretized version would give readers a more complete picture of practical performance.

---

> ### Author Rebuttal · Authors · 2025-07-30
>
> We thank the reviewer for the careful review, and we’re happy to answer the questions raised by the reviewer.
>
> 1. On the binary group features
>
>     As pointed out by the reviewers, LSBoost extends the binary groups to a function class that is closed under affine transformations, while currently, we do not fully support such a function class. This extends the scope of our algorithm beyond group fairness, and we intend to leave this for future investigation. We’ll also discuss more about this in point 3.
>
> 2. On the evaluation in section 5.3.2
>
>     We’d like to point out that the metric we use in 5.3.2 is the Multicalibration Error in Definition 3.2, which only works with discretized predictors. Therefore, it’s not possible to directly calculate this metric using the undiscretized version of the predictor. Indeed in practice, there may be other metrics that practitioners might be interested in, and we provided a smooth indicator of multicalibration in Section 5.3.3. For other practical metrics beyond multicalibration that works on continuous outputs, we reported the accuracy and loss in our response to Reviewer zBZu, and we kindly refer you to our answer there.
>
> 3. On the behaviors of hidden groups, and extensions to deeper trees
>
>     This is a great question! While we cannot guarantee performance on general unspecified "hidden" groups, it might be possible to extend our approach if these groups are assumed to have certain structures, which include:
>
>     - Intervals on continuous features. Trees can learn to take thresholds on continuous features, and two trees of depth one on some feature can represent an interval on this feature. If we assume that there are some unknown sensitive groups that can be represented by intervals on some continuous feature, ensembles of trees of depth 2 could still learn to be multicalibrated wrt to that group.
>     - Intersections on identifiable groups. If hidden groups are intersections of existing binary indicators or continuous feature intervals, then using trees of depth three instead (one level for $f_0$, two for group intersections) could enable multicalibration for these composite groups.
>
>     Note that the tree ensemble will try to be multicalibrated w.r.t. all possible groups in this manner. For instance, the continuous age feature can be passed if we want the model to be multicalibrated to all age intervals. It's important to note that explicit group definition is generally preferable when possible, because extending to these complex structures increases model complexity, potentially leading to decreased performance due to bias-variance trade-offs. For instance, using depth-3 trees requires multicalibration with respect to all sets described by intersections of groups, which is a much more challenging task.
>
> 4. On the empirical details of the number of trees
>
>    We evaluate the number of trees of our best performing model on each task, averaged over 10 folds, and present the result in the column “Ours”.
>
>    As for LSBoost, in most of the cases, trees of depth 1 performs better than trees of depth 2 as a weak learner, and these cases are removed from the evaluation. We record the number of refinement rounds of LSBoost as well as the total number of trees used in the remaining cases, and present the results in the column "LSBoost".
>
>    Table: Number of trees used for each task.
>
>    |                                   | Ours           | LSBoost                                              |
>    |:----------------------------------|:---------------|:-----------------------------------------------------|
>    | Skin Lesion Classification        | 423.9 ± 97.5   |                                                      |
>    | Income Classification             | 830.8 ± 133.4  | #Grid = 10: 64.2 ± 13.5 trees in 17.6 ± 4.5 rounds   |
>    |                                   |                | #Grid = 30: 441.9 ± 83.1 trees in 58.5 ± 16.0 rounds |
>    | Comments Toxicity Classification  | 558.0 ± 103.3  |                                                      |
>    | Age Regression                    | 2496.1 ± 677.5 | #Grid = 20: 94.9 ± 14.0 trees in 8.9 ± 1.4 rounds    |
>    |                                   |                | #Grid = 30: 90.3 ± 30.6 trees in 7.0 ± 1.9 rounds    |
>    | Income Regression                 | 1201.3 ± 171.4 |                                                      |
>    | Travel Time Regression            | 108.2 ± 48.7   | #Grid = 10: 36.0 ± 32.2 trees in 15.0 ± 12.6 rounds  |
>    |                                   |                | #Grid = 30: 11.6 ± 6.0 trees in 6.2 ± 3.0 rounds     |
>    |                                   |                | #Grid = 100: 8.0 ± 3.8 trees in 4.6 ± 2.8 rounds     |
>
>    The number of trees actually varies a lot across datasets, but in general, our models consist of more trees.

---

> > ### Comment · Reviewer_3K17 · 2025-08-08
> >
> > Thank you for your comments and the additional experimental details. I have raised my score to 5.

---

### Note · Authors · 2025-08-16

We are grateful for the reviewers' thorough evaluation and constructive feedback. Through the rebuttal process, we have addressed the concerns raised by the reviewers, which motivate us to make the following changes to our paper to minimize the potential confusion for the general audience.

- **Clarifications on the details of algorithm implementation:** Reviewer kSyY pointed out an inconsistency between Fig. 1 and Eq. 4. We agree that this might cause confusion in understanding our algorithm, and we will revise Fig. 1 for consistency. We will also set aside a separate section in the appendix to include our “Further explanation (1/3)” to kSyY to explain why our formulation is equivalent to mainstream implementation of decision tree ensemble solvers. We will also correct the inaccurate notation in Eq. 3.

- **Intuitions of our algorithm:** In our discussions with Reviewer kSyY, we pointed out that even if the group indicators were derived from original features, i.e., “no new information about the underlying data distribution is gained”, our algorithm can still provide substantial multicalibration improvements through enhanced representation power and targeted feature engineering for group fairness. This provides an intuitive understanding of our algorithm, and we’d like to include it in our introduction.

- **Additional Empirical Study:** We conducted additional empirical evaluations, as presented in our answer to the reviewers. We find some reviewers (3K17, zBZu) interested in metrics other than multicalibration, and we agree that loss and accuracy can provide practitioners with comprehensive performance insights. We would include these additional evaluations in our main body and/or appendix.

- **Motivations, Limitations, and Future Directions:** We agree with Reviewer Wact that our motivation would benefit from additional citations, and we will add it to our introduction part. Some reviewers (zBZu, 3K17) also discussed with us the limitations and possible future directions. Currently, our algorithm is limited to binary group features, but this could be extended to include more flexible group structures, as indicated by our third point in our rebuttal to Reviewer 3K17. We believe it would be beneficial to incorporate such discussions in our concluding section for directions of future work.

Again, we thank the reviewers for their valuable feedback, which allows us to further improve our paper.

---

### Decision · Program_Chairs · 2025-09-17

**Decision:**

Accept (poster)

**Comment:**

This paper proposes a discretization-free approach to multicalibration based on optimizing a loss objective over ensembles of depth-two decision trees. The work is motivated by the limitations of prior discretization-based methods, which introduce rounding errors, additional hyperparameters, and potential distortions in downstream use. The authors provide theoretical guarantees under a "loss saturation" condition, and present empirical evaluations across multiple datasets showing improvements over existing baselines.

The reviewers found the paper to be technically solid, novel, and relevant to the community. They noted that the method elegantly eliminates the discretization bottleneck, while retaining multicalibration guarantees. The approach is simple to implement (using standard tree ensemble solvers) and empirically performs competitively or better than previous multicalibration methods, especially on worst-group calibration metrics. The theoretical contribution was found to be well-motivated and rigorous. Reviewers also appreciated the clear presentation of the results.